# Dense Associative Memory with Epanechnikov Energy

**Benjamin Hoover**
IBM Research
Georgia Tech

**Zhaoyang Shi**
Harvard

**Krishnakumar Balasubramanian**
UC Davis

**Dmitry Krotov**
IBM Research

**Parikshit Ram**
IBM Research

## Abstract

We propose a novel energy function for Dense Associative Memory (DenseAM) networks, the log-sum-ReLU (LSR), inspired by optimal kernel density estimation. Unlike the common log-sum-exponential (LSE) function, LSR is based on the Epanechnikov kernel and enables exact memory retrieval with exponential capacity without requiring exponential separation functions. Moreover, it introduces abundant additional *emergent* local minima while preserving perfect pattern recovery — a characteristic previously unseen in DenseAM literature. Empirical results show that LSR energy has significantly more local minima (memories) that have comparable log-likelihood to LSE-based models. Analysis of LSR's emergent memories on image datasets reveals a degree of creativity and novelty, hinting at this method's potential for both large-scale memory storage and generative tasks.

## 1 Associative Memories and Energy Landscapes

Energy-based Associative Memory networks or AMs are models parameterized with $M$ "memories" in $d$ dimensions, $\Xi = \{\boldsymbol{\xi}_\mu \in \mathbb{R}^d, \mu \in [\![M]\!]\}$. A popular class of models from this family can be described by an energy function defined on the state vector $\mathbf{x} \in \mathcal{X} \subseteq \mathbb{R}^d$:

$$E_\beta(\mathbf{x}; \Xi) = -Q\left[\sum_{\mu=1}^{M} F\left(\beta S\left(g(\mathbf{x}), \boldsymbol{\xi}_\mu\right)\right)\right], \tag{1}$$

where $g : \mathbb{R}^d \to \mathbb{R}^d$ is a vector operation (such as binarization, (layer) normalization), $S : \mathbb{R}^d \times \mathbb{R}^d \to \mathbb{R}$ is a similarity function (e.g., dot-product, negative Euclidean distance), $\beta > 0$ denotes the inverse temperature, $F : \mathbb{R} \to \mathbb{R}$ is a rapidly growing separation function (power, exponential) and $Q$ is a monotonic scaling function (logarithm, linear) [1, 2, 3]. With $g$ as the sign-function, $\boldsymbol{\xi}_\mu \in \{-1, +1\}^d$, $S(\mathbf{x}, \mathbf{x}') = \langle \mathbf{x}, \mathbf{x}' \rangle$ and $F$ as the quadratic function, and $Q$ as a linear function, we recover the classical Hopfield model [4]. The output of an associative memory (AM) corresponds to one of the local minima of its energy function. A memory $\boldsymbol{\xi}_\mu$ is said to be retrieved if $\mathbf{x} \approx \boldsymbol{\xi}_\mu$ corresponds to such a local minimum, with exact retrieval occurring when $\mathbf{x} = \boldsymbol{\xi}_\mu$. The memory capacity of the AM is defined as the maximum number $M^\star$ of correctly retrieved memories. For classical AMs, the capacity scales as $M^\star = O(d)$. With the introduction of power-law separation functions — that is, $F(x) = x^p$ for $p > 2$ — modern *Dense Associative Memories* (DenseAMs) achieve a significantly higher capacity of $M^\star = O(d^p)$ [1, 5].

The use of an exponential separation function combined with a logarithmic scaling function — $F(x) = \exp(x)$ and $Q(x) = \log x$ — leads to the widely studied log-sum-exp (LSE) energy function [6, 7, 8], yielding exponential memory capacity $M^\star \sim \exp(d)$ [9]. Hierarchical organization of memories has also been explored in Krotov [10] and Hoover et al. [11]. Given that the gradient

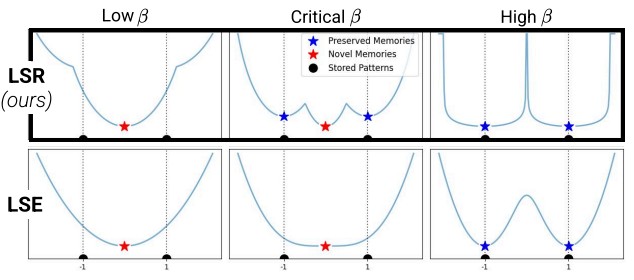

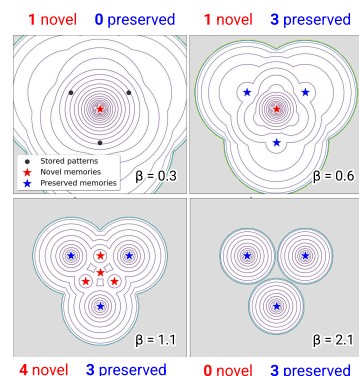

Figure 1: LSR energy can create more memories than there are stored patterns under critical regimes of $\beta$. Left: 1D LSR vs LSE energy landscape. Note that LSE is never capable of having more local minima than the number of stored patterns. Right: 2D LSR energy landscape, where increasing $\beta$ creates novel local minima where basins intersect. Unsupported regions are shaded gray.

of the LSE energy corresponds to a softmax over all stored patterns, recent works have proposed sparsified variants to improve scalability. In particular, Hu et al. [12] and dos Santos et al. [13] consider sparsified softmax-based gradients, effectively projecting the full gradient onto a reduced support.[1] Alternatively, Wu et al. [14] propose learning new representations for the memories to increase capacity, while continuing to use the LSE energy in the transformed representation space.

In this work, we consider the following motivating question: *can we achieve simultaneous perfect memorization and generalization in associative memory models?* While the exponential separation function enables DenseAMs to achieve high memory capacity, capacity alone is not the only desideratum. In standard supervised learning, it was long believed that exactly interpolating or memorizing the training data — achieving zero training loss — would harm generalization. However, recent advances, particularly in deep learning, have challenged this belief: models that perfectly fit the training data can still generalize well. Although this phenomenon gained prominence with deep networks, it has earlier roots in kernel methods and boosting [15, 16, 17].

An analogous question arises in the context of associative memory (AM) models. Traditionally, AMs focus on storing a fixed set of patterns. But from a broader machine learning perspective, the goal extends beyond memorization to include the generation of new, meaningful patterns. Prior work using LSE-type energy functions has shown that generating such novel patterns typically requires sacrificing perfect recall of the original patterns. This trade-off highlights a core tension between memorization and generalization. To address this, we explore alternative separation functions that can preserve exact memorization while enabling the emergence of new patterns — pushing toward models that truly unify memory and generalization.

Our approach is also motivated by the well-established connection between the energy and probability density function. An energy function $\mathsf{E} : \mathbb{R}^d \to \mathbb{R}$ induces a probability density function $p : \mathbb{R}^d \to \mathbb{R}_{\geq 0}$ with $p(\mathbf{x}) = \exp[-\mathsf{E}(\mathbf{x})]/\int_{\mathbf{z}} \exp[-\mathsf{E}(\mathbf{z})]d\mathbf{z}$. Conversely, given a density $p$, we have an energy $\mathsf{E}(\mathbf{x}) \propto -\log p(\mathbf{x})$, the negative log-likelihood. Minimizing the energy corresponds to maximizing the log-likelihood (with respect to the corresponding density). Based on this connection, with $Q(\cdot) = \log(\cdot)$, the $\exp[-E_\beta(\mathbf{x}; \boldsymbol{\Xi})] = \sum_\mu F(\beta S(\mathbf{x}, \boldsymbol{\xi}_\mu))$ in eq. (1) (assuming $g$ is identity) is the corresponding (unnormalized) density at $\mathbf{x}$. Assuming that the memories $\boldsymbol{\xi}_\mu \sim p$ are sampled from an unknown ground truth density $p$, the $\exp[-E_\beta(\mathbf{x}; \boldsymbol{\Xi})]$ is an unnormalized **kernel density estimate** or KDE of $p$ at $\mathbf{x}$ with the *kernel $F$* and bandwidth $1/\beta$ [18]. Thus, the LSE energy with $F(x) = \exp(x)$ and $S(\mathbf{x}, \mathbf{x}') = -1/2\|\mathbf{x} - \mathbf{x}'\|^2$ corresponds to the KDE of $p$ with the Gaussian kernel.

KDE is well studied in nonparametric statistics [18, 19], and various forms of kernels have been explored. The quality of the estimates are well characterized in terms of properties on the kernels; we will elaborate on this in the sequel. While the Gaussian kernel is extremely popular for KDE (much like LSE in AM literature), there are various other kernels which have better estimation abilities than the Gaussian kernel. Among the commonly used kernels, the Epanechnikov kernel

---

[1]Sparsified softmax-based gradients can be interpreted as specific projections of the original gradient.

has the most favorable estimation quality (see section 2). In our notation, this corresponds to a kernel $F(x) = \max(1 + x, 0) = \text{ReLU}(1 + x)$, a shifted ReLU operation (again with $S(\mathbf{x}, \mathbf{x}') = -1/2\|\mathbf{x} - \mathbf{x}'\|^2$). This results in a novel energy function that we name log-sum-ReLU or LSR (see eq. (3)). *Surprisingly, we show that this energy function is capable of both exactly memorizing all $M$ original patterns (with $M \sim \exp(d)$) and simultaneously generating new, meaningful emergent memories* (see Definition 2). This defies the conventional tradeoff seen in prior AM models — where improving generalization (ability to create emergent local minima) typically requires compromising exact memorization — and reveals that precise memory storage and creative pattern generation are not inherently at odds with one another. To summarize, we make the following contributions in this work:

- **Novel ReLU-based energy function with exponential memory capacity.** We propose a LSR energy function for DenseAM using the popular ReLU activation, built upon the connection between energy functions and densities. In Theorems 1 and 2 respectively, we demonstrate exact retrieval and exponential memory capacity of LSR energy, without the use of $\exp(\cdot)$ separation function.
- **Simultaneous memorization and emergence.** We show that this LSR energy has a unique property of *simultaneously* being able to *exactly* retrieve all original memories (training data points) while also creating many *emergent* memories (additional local minima). The total number of memories of LSR can exceed the number of stored patterns, a property absent with LSE (see Proposition 1). When applied to images, LSR can generate novel and seemingly creative memories that are not present in the training dataset.

## 2 Kernel Density Estimation and the Choice of Kernels

We now provide a brief overview of Kernel Density Estimation (KDE) considering the univariate setting for simplicity; similar conclusions also hold in higher dimensions. Given a sample $\boldsymbol{\Xi} = \{\xi_\mu \in \mathbb{R}, \mu \in [\![M]\!]\}$ drawn from an unknown density $f$, the KDE is defined as $\hat{f}_h(\xi) = (Mh)^{-1} \sum_{\mu=1}^M K\left(\frac{\xi - \xi_\mu}{h}\right)$, where $K(\cdot)$ is the kernel function and $h > 0$ is the bandwidth parameter. The kernel function is assumed to satisfy: (i) symmetry (i.e., $K(-x) = K(x)$, for all $x \in \mathbb{R}$), (ii) positivity (i.e., $K(x) \geq 0$, for all $x \in \mathbb{R}$) and (iii) normalization (i.e., $\int_x K(x)\, dx = 1$). Note that for the purpose of KDE, the scale of the kernel function is not unique. That is, for a given $K(\cdot)$, we can define $\tilde{K}(\cdot) = b^{-1} K(\cdot/b)$, for some $b > 0$. Then, one obtains the same KDE by rescaling the choice of $h$. Hence, the shape of the kernel function plays a more important role in determining the choice of the kernel. We now introduce two parameters associated with the kernel, $\mu_K := \int_x x^2 K(x)\, dx$ and $\sigma_K := \int_x K^2(x)\, dx$ that correspond to the *scale* and *regularity* of the kernel. We will discuss below how the *generalization error* of KDE depends on the aforementioned parameters.

The *generalization error* of $\hat{f}_h(\xi)$ is measured by the Mean Integrated Squared Error (MISE), given by $\mathsf{MISE}(h) = \mathbb{E}\left[\int_\xi (\hat{f}_h(\xi) - f(\xi))^2 d\xi\right]$. Assuming that the ground-truth density $f(\xi)$ is twice continuously differentiable, a second-order Taylor expansion gives the leading terms of the $\mathsf{MISE}(h)$, which decomposes into squared bias and variance terms: $\mathsf{MISE}(h) \approx \frac{\mu_K^2}{4} h^4 \int_\xi |f''(\xi)|^2 d\xi + \frac{\sigma_K}{Mh}$; see Wand and Jones [18, Section 2.5] for details. This result shows that reducing $h$ decreases bias but increases variance, while increasing $h$ smooths the estimate but introduces bias, highlighting the bias-variance trade-off. The optimal mean-square is obtained by minimizing $\mathsf{MISE}(h)$ with respect to $h$. We thus obtain the optimal choice of $h$ and the optimal generalization accuracy as

$$h_* := \left(\frac{\sigma_K}{M\mu_K^2} \frac{4}{\int_\xi |f''(\xi)|^2 d\xi}\right)^{1/5} \quad \text{and} \quad \mathsf{MISE}(h_*) \approx \frac{5}{4}\left(\frac{\sqrt{\mu_K}\sigma_K \int_\xi |f''(\xi)|^2 d\xi}{M}\right)^{4/5}, \quad (2)$$

respectively. From this, we see that the choice of the kernel $K$ in the KDE, controls the generalization error via the term $\sqrt{\mu_K}\sigma_K$.

Thus, a natural question is to find the choice of kernel $K(\cdot)$ that results in the minimum $\mathsf{MISE}(h_*)$. As discussed above, the scale of the kernel function is non-unique. Hence, the problem boils down to minimizing $\sigma_K$ (which is regularity parameter of the kernel, determining the shape), subjected to $\mu_K = 1$ (without loss of generality), over the class of normalized, symmetric, and positive kernels.

This problem is well-studied (see, for example, [20], [21], Wand and Jones [18, Section 2.7]), and, as it turns out, the Epanechnikov kernel $K_{\text{epan}}(x) = \max\{1 - x^2, 0\} = \text{ReLU}\left(1 - x^2\right)$ achieves the optimal *generalization error*. The quantity, $\text{Eff}(\mathsf{K}) \coloneqq \sigma_K / \sigma_{K_{\text{epan}}}$ is hence referred to as the efficiency of any kernel with respect to the Epanechnikov kernel. A description of choices for kernel functions and their efficiency relative to the Epanechnikov kernel is provided in Appendix C.

The number of modes of the KDE has also been examined in the literature, mostly when the target is unimodal. Assuming that the target is unimodal, a direct consequence of Mammen [22, Theorem 1] on the number of modes of the KDE when $d = 1$ (also see Geshkovski et al. [23, Theorem 1.1]) is that the number of modes of KDE with a Gaussian kernel with bandwidth $h$ is $\widetilde{\Theta}(1/\sqrt{h})$; see Geshkovski et al. [23, Section 1.2] for extensions to dimension $d > 1$.

# 3 A New Energy Function with Emergent Memory Capabilities

So far, we have seen the relationship between the LSE energy and the KDE, i.e., $\exp[-E_\beta(\mathbf{x}; \boldsymbol{\Xi})]$ is an unnormalized kernel density estimate with the Gaussian kernel and the bandwidth $1/\beta$, and the optimality of using the Epanechnikov kernel in KDE. Given these observations, we will explore the use of the corresponding shifted-ReLU separation function $\text{ReLU}\left(1 + x\right)$ in the energy function instead of the widely used exponentiation. Before we state the precise energy functions, we compare and contrast the shapes of these separation functions $F(\beta x)$ in fig. 2 for varying values of the inverse temperature $\beta$. Note that, as the $\beta$ increases, both these separation functions decay faster. However, as expected, the shifted-ReLU separation linearly decays and then zeroes out.

Recall that LSE ENERGY is given by $E_\beta^{\text{LSE}}(\mathbf{x}; \boldsymbol{\Xi}) = -\frac{1}{\beta} \log \sum_{\mu=1}^M \exp(-\frac{\beta}{2} \|\mathbf{x} - \boldsymbol{\xi}_\mu\|^2)$. Based on the discussion on separation functions, our proposed LSR ENERGY (which we also refer to as Epanechnikov energy) is given by

$$E_\beta^{\text{LSR}}(\mathbf{x}; \boldsymbol{\Xi}) = -\frac{1}{\beta} \log \left( \epsilon + \sum_{\mu=1}^M \text{ReLU}\left( 1 - \frac{\beta}{2} \|\mathbf{x} - \boldsymbol{\xi}_\mu\|^2 \right) \right), \tag{3}$$

where $\|\cdot\|$ describes the Euclidean norm and $\beta$ is an inverse temperature.

The factor $\epsilon \geq 0$ in the LSR energy is a small non-negative constant, where an $\epsilon > 0$ ensures that every point in the space has finite (albeit extremely large $O(\log(1/\epsilon))$) energy for all values of $\beta$. Indeed, with $\epsilon = 0$, defining $\mathsf{S}_\mu \triangleq \{\mathbf{x} \in \mathcal{X} : \|\mathbf{x} - \boldsymbol{\xi}_\mu\| \leq \sqrt{2/\beta}\}$, it is easy to see that $\forall \mathbf{x} \in \mathcal{X} \setminus \cup_{\mu=1}^M \mathsf{S}_\mu$, $E_\beta^{\text{LSR}}(\mathbf{x}) = \infty$. This is a result of the finite-ness of the ReLU separation function. Regions of infinite energy implies zero probability density, which matches the finite support of the density estimate with the Epanechnikov kernel. Based on the intro-

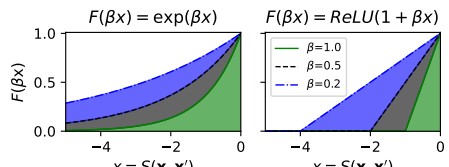

Figure 2: Visualizing the separation functions $F(\beta x) = \exp(\beta x)$ (LSE) and $F(\beta x) = \text{ReLU}\left(1 + \beta x\right)$ (LSR) with $x = S(\mathbf{x}, \mathbf{x}')$ for varying values of $\beta$. We focus on $S(\mathbf{x}, \mathbf{x}') = -\frac{1}{2}\|\mathbf{x} - \mathbf{x}'\|^2$.

duced LSR energy, we next highlight the following favourable properties; see appendix E for the proofs and technical details.

**Theorem 1.** *Let $r = \min_{\mu, \nu \in [\![M]\!], \mu \neq \nu} \|\boldsymbol{\xi}_\mu - \boldsymbol{\xi}_\nu\|$ be the minimum Euclidean distance between any two memories. Let $S_\mu(\Delta) = \{\mathbf{x} \in \mathcal{X} : \|\mathbf{x} - \boldsymbol{\xi}_\mu\| \leq \Delta\}$ be a basin around the $\mu^{th}$ memory for some basin radius $\Delta \in (0, r)$. Then, with $\beta = 2/(r - \Delta)^2$, for any $\mu \in [\![M]\!]$ and any input $\mathbf{x} \in S_\mu(\Delta)$, the output of the DenseAM via LSR energy gradient descent is exactly $\boldsymbol{\xi}_\mu$, implying that all memories $\boldsymbol{\xi}_\mu, \mu \in [\![M]\!]$ are retrievable. Furthermore, if the learning rate of the energy gradient descent is set appropriately, then for any $\mu \in [\![M]\!]$ and any $\mathbf{x} \in S_\mu(\Delta)$, the memory is exactly retrieved with a single energy gradient descent step (single step retrieval).*

The above result states that, given a set of memories, and an appropriately selected $\beta$, there is a distinct basin of attraction $S_\mu(\Delta)$ around each memory $\boldsymbol{\xi}_\mu$, and any input $\mathbf{x}$ from within that basin exactly retrieves the memory as the output of the DenseAM.

**Remark 1.** *For a finite but appropriately large $\beta$, the LSR energy gradient $\nabla E_\beta^{LSR}(\boldsymbol{\xi}_\mu; \boldsymbol{\Xi})$ at any memory $\boldsymbol{\xi}_\mu$ is exactly zero, implying exact retrieval of the memory. The LSE energy gradient $\nabla E_\beta^{LSE}(\boldsymbol{\xi}_\mu; \boldsymbol{\Xi})$ is only approximately zero, and the retrieved point is approximately equal to an*

*original memory [7, Theorem 3]. However, if $\beta = \infty$ then the LSE energy gradient is exactly zero at the memory.*

The striking phenomenon that we observe with LSR energy is that the DenseAM can simultaneously create local energy minima around the original memories as well as additional local minima around points that are not part of the set of original memories; see fig. 1. We formalize this concept below through the notion of *global emergence*.

**Definition 1** (Novel local minima). *Consider a DenseAM parameterized with $M$ memories $\Xi = \{\boldsymbol{\xi}_1, \ldots, \boldsymbol{\xi}_M\}, \boldsymbol{\xi}_\mu \in \mathcal{X}$, and equipped with an energy function $E_\beta(\mathbf{x}; \Xi)$ at any state $\mathbf{x} \in \mathcal{X}$ for a specific inverse temperature $\beta > 0$. For some $\varepsilon > 0$, we define $\mathcal{M}_\varepsilon$ as the (possibly empty) set of **novel local minima** $\tilde{\boldsymbol{\xi}} \in \mathcal{X}$ such that $\forall \tilde{\boldsymbol{\xi}} \in \mathcal{M}_\varepsilon$,*

*(a) $\tilde{\boldsymbol{\xi}}$ is a local energy minimum with $\nabla E_\beta(\tilde{\boldsymbol{\xi}}; \Xi) = 0$ and $\nabla^2 E_\beta(\tilde{\boldsymbol{\xi}}; \Xi) \succ 0$,*

*(b) $\tilde{\boldsymbol{\xi}}$ is novel with respect to the original memories, that is, $\min_{\mu \in [\![M]\!]} \left\| \tilde{\boldsymbol{\xi}} - \boldsymbol{\xi}_\mu \right\| \geq \varepsilon$.*

**Definition 2** (Global emergence). *For the DenseAM in Definition 1 and for some $\varepsilon > 0$, let $\mathcal{M}_\varepsilon$ be the (possibly empty) set of novel local minima. For some $\beta \subset (0, \infty)$, we claim that this system exhibits $\varepsilon$-**global emergence** if (i) each original memory $\boldsymbol{\xi}_\mu, \mu \in [\![M]\!]$ is a local energy minimum with $\nabla E_\beta(\boldsymbol{\xi}_\mu; \Xi) = 0$ and $\nabla^2 E_\beta(\boldsymbol{\xi}_\mu; \Xi) \succ 0$ (positive definite), and (ii) the set $\mathcal{M}_\varepsilon$ is non-empty. We term $\mathcal{M}_\varepsilon$ as the set of $\varepsilon$-**globally emergent memories**.*

The notion of $\varepsilon$-global emergence specifically refers to those new patterns that arise *after* all original memories have been exactly stored. Definition 2 characterizes emergence as a property of the global energy function at a specific inverse temperature, requiring simultaneous exact recovery of all original memories and the presence of at least one novel local minimum (parameterized with $\varepsilon$). It is instructive to start by understanding the above definition for DenseAMs equipped with LSE energy. According to Ramsauer et al. [7], any point $\mathbf{x}^*$ such that $\nabla E_\beta^{\text{LSE}}(\mathbf{x}^*; \Xi) = 0$ is defined via the soft-max corresponding to the transformer attention as follows: $\mathbf{x}^* = \sum_{\mu=1}^M \text{softmax}(\beta(\mathbf{x}^*)^\top \boldsymbol{\xi}_\mu) \boldsymbol{\xi}_\mu$, and the softmax can be highly peaked if all $\{\boldsymbol{\xi}_\mu\}_{\mu=1}^M$ are well separated and $\mathbf{x}^*$ is near a stored pattern $\boldsymbol{\xi}_\mu$. If no stored pattern $\boldsymbol{\xi}_\mu$ is well separated from the others, then $\mathbf{x}^*$ is close to a global fixed point, which is the arithmetic mean of all the stored patterns. Based on this, we can make the following observations:

- Case I: *All LSE memories are novel.* With a large enough but finite $\beta$, there is a minimum close to each of the original memories. However, each of these local minima will be considered novel local minima as these are distinct from the original memories, thus condition (ii) in Definition 2 will be satisfied. However, then the condition (i) in Definition 2 would not be satisfied.
- Case II: *No LSE memories are novel.* If we do consider the case $\beta = \infty$, then the original memories would exactly be the local energy minima, and condition (i) will be satisfied. But then the set of novel local minima $\mathcal{M}_\varepsilon$ for a strictly positive $\varepsilon$ would be empty, violating condition (ii).
- Case III: *Novel LSE memories form only when basins merge.* For a moderate $\beta$, LSE can form novel local minima by merging the basins of attractions of the original memories, thereby giving us a non-empty $\mathcal{M}_\varepsilon$ and satisfying condition (ii) in Definition 2. However, condition (i) will be violated as the memories whose basins are merged would no longer be local minima.

Thus, we can make this more formal in the following:

**Proposition 1.** *Assume $\{\boldsymbol{\xi}_\mu\}_{\mu=1}^M$ are i.i.d. from any density fully supported on $\mathcal{X}$. Note that they are linearly independent with probability 1, as otherwise they lie in a lower dimensional space. Then, for any $\beta > 0$, the LSE energy, $E_\beta^{LSE}(\cdot)$, does not satisfy the $\varepsilon$-global emergence in Definition 2.*

One can argue that global emergence as in Definition 2 is too restrictive; we also want to characterize an individual local minimum as "emergent", or not. Thus we present a relaxed *local* notion of emergence in the following, noting that LSE does not satisfy this weaker form of emergence either:

**Definition 3** (Locally emergent memory). *Consider the DenseAM in Definition 1 with a non-empty set of novel local minima $\mathcal{M}_\varepsilon$ for a $\varepsilon > 0$. For any $\tilde{\boldsymbol{\xi}} \in \mathcal{M}_\varepsilon$, let $\mathcal{S}(\tilde{\boldsymbol{\xi}}) \subseteq \Xi$ be the minimal non-empty subset of $\Xi$ such that, for each $\boldsymbol{\xi}_\mu \in \mathcal{S}(\tilde{\boldsymbol{\xi}})$, $\tilde{\boldsymbol{\xi}}$ is no longer a local minimum of the energy $E_\beta(\cdot, \Xi \setminus \{\boldsymbol{\xi}_\mu\})$ that excludes the memory. Then we define $\tilde{\boldsymbol{\xi}} \in \mathcal{M}_\varepsilon$ as a $\varepsilon$-**locally emergent memory** if there is some original memory $\boldsymbol{\xi}_\mu \in \mathcal{S}(\tilde{\boldsymbol{\xi}})$ which still is a local minimum of the energy*

$E_\beta(\cdot; \Xi)$. *If every original memory $\xi_\mu \in \mathcal{S}(\tilde{\xi})$ is a local minimum of $E_\beta(\cdot; \Xi)$, we call $\tilde{\xi} \in \mathcal{M}_\varepsilon$ a* $\varepsilon$**-local strongly emergent memory**.

While Definition 2 discusses emergence at a global energy level, Definition 3 characterizes emergence locally for each of the novel local minima. This distinction is important as we see emergence as a general property of the energy function that can be driven by a subset of the memories. Definition 2 requires all original memories to be retrievable, while in Definition 3 we allow for emergence due to the interaction of stored patterns in a system even if not all original memories are retrievable, so long as a critical subset of the original memories are. Global emergence implies the existence of at least one local strongly emergent memory; every globally emergent memory is a local strongly emergent one. However, the existence of a local strongly emergent memory does not imply global emergence. As with global emergence, we can see that a DenseAM with LSE energy does not also have locally emergent memories. First note that, for a finite $\beta$, all local minima are novel local minima, and the minimal set $\mathcal{S}(\tilde{\xi})$ for a novel local minimum $\tilde{\xi}$ is the whole set $\Xi$ given the infinite support of the exponential function, with none of them being a local minimum. For $\beta = \infty$, the set of novel local minima $\mathcal{M}_\varepsilon$ is empty. So in both cases, the required conditions are not satisfied and there are no locally emergent memories with LSE energy.

Note that these novel local minima are different from the well-studied spurious memories or parasitic memories [24, 25]. In classical AMs, spurious memories start appearing when the AM is packed with memories beyond its memory capacity. In contrast, the appearance of emergence (novel local minima) does not seem to be related to whether the DenseAM is over or under capacity — as we show in fig. 1, a locally emergent memory can appear even with just 2 stored patterns of any dimension. Spin-glass states [26] do not occur in either the LSE or LSR energy due to our use of Euclidean similarity over the dot product in both energies.

The next result provides explicit characterization of the form of novel memories in LSR energy.

**Proposition 2.** *Consider the LSR energy in eq. (3). For any $\mathbf{x} \in interior(\mathcal{X})$, letting $B(\mathbf{x}) \triangleq \{\mu \in [\![M]\!] : \|\mathbf{x} - \xi_\mu\| \leq \sqrt{2/\beta}\}$, there is a local minima of the LSR energy which is given by* $\frac{1}{|B(\mathbf{x})|} \sum_{\mu \in B(\mathbf{x})} \xi_\mu$.

Note that when $|B(\mathbf{x})| = 1$, the local minima in Proposition 2 is exactly the stored memory $\{\xi_\mu\}_{\mu=1}^M$. With $|B(\mathbf{x})| > 1$, it is not equal to any of the original memories $\{\xi_\mu, \mu \in [\![M]\!]\}$ (with probability 1). The region $\{\mathbf{x} \in \mathcal{X} : |B(\mathbf{x})| > 1\} \subset \mathcal{X}$ is precisely characterized as $\left( \cup_{\mu \in [\![M]\!]} \mathsf{S}_\mu \right) \setminus \left( \cup_{\mu \in [\![M]\!]} S_\mu(\Delta) \right)$ where $\mathsf{S}_\mu$ is the region of finite energy around the $\mu^{\text{th}}$ memory and $S_\mu(\Delta)$ (defined in Theorem 1) is the distinct attracting basin for the $\mu^{\text{th}}$ memory. The following theorem shows that this LSR based DenseAM is capable of simultaneously retrieving all (up to exponentially many) memories while also creating many novel local minima, and quantifies this phenomenon precisely.

**Theorem 2.** *Consider a DenseAM parameterized with $M$ memories $\Xi = \{\xi_1, \ldots, \xi_M\}$ sampled uniformly from $\mathcal{X}$ with $vol(\mathcal{X}) = V < \infty$ and the LSR energy $E_\beta^{LSR}(\cdot; \Xi)$ defined in eq. (3). For each novel local minimum $\mathbf{x}^* = \frac{1}{|B(\mathbf{x}^*)|} \sum_{\mu \in B(\mathbf{x}^*)} \xi_\mu$ given in Proposition 2, define $D_{\max}(\mathbf{x}^*) := \max_\mu \|\mathbf{x}^* - \xi_\mu\|$,*

$$\delta_{\min}(\mathbf{x}^*) := \min_{\mu \in B(\mathbf{x}^*)} \left( \frac{2}{\beta} - \|\mathbf{x}^* - \xi_\mu\|^2 \right), \quad \gamma_{\min}(\mathbf{x}^*) = \min_{\nu \notin B(\mathbf{x}^*)} \left( \|\mathbf{x}^* - \xi_\nu\|^2 - \frac{2}{\beta} \right).$$

*Then, for all $\beta > 0$ such that $\max_{\mathbf{x}^*} \delta_{\min}(\mathbf{x}^*) > 0$ and $\min_{\mathbf{x}^*} \gamma_{\min}(\mathbf{x}^*) > 0$, there exists an $\varepsilon := \min_{\mathbf{x}^*} \left( \sqrt{D_{\max}^2(\mathbf{x}^*) + \min\{\delta_{\min}(\mathbf{x}^*), \gamma_{\min}(\mathbf{x}^*)\}} - D_{\max}(\mathbf{x}^*) \right) > 0$ such that $E_\beta^{LSR}(\cdot)$ satisfies the $\varepsilon$-global emergence condition (Definition 2) with high probability, as:*

(a) *With probability at least $\delta \in (0,1)$, and $M = \Theta\left( \sqrt{1-\delta} \exp(\alpha d) \right)$ for a positive $\alpha$, all memories are retrievable as per Theorem 1 with the value of the minimum pairwise distance $r = \min_{\mu, \mu \in [\![M]\!], \mu \neq \nu} \|\xi_\mu - \xi_\nu\| \geq (V_d/V)^{-1/d} e^{-2\alpha}$ and per-memory basin radius $\Delta \in (0, (V_d/V)^{-1/d} e^{-2\alpha})$ with a $\beta \leq 2/((V_d/V)^{-1/d} e^{-2\alpha} - \Delta)^2$, where $V_d$ is the volume of the unit ball in $\mathbb{R}^d$.*

(b) *For each novel local minimum $\mathbf{x}^*$, there exists a radius*

$$r^* := \sqrt{D_{\max}^2(\mathbf{x}^*) + \min\{\delta_{\min}(\mathbf{x}^*), \gamma_{\min}(\mathbf{x}^*)\}} - D_{\max}(\mathbf{x}^*) > 0, \tag{4}$$

*such that $S_{\mathbf{x}^*}(r^*) = \{\mathbf{x} \in \mathcal{X} : \|\mathbf{x} - \mathbf{x}^*\| \leq r^*\}$ forms a basin around the novel memory $\mathbf{x}^*$, and for any $\mathbf{x} \in S_{\mathbf{x}^*}(r^*)$, the output of the DenseAM via energy gradient descent is exactly $\mathbf{x}^*$, implying the novel memories are retrievable.*

*Furthermore, with probability at least $1 - M^{-2}$, the number of $\varepsilon$-globally emergent memories is*

$$O\left(\exp\left(M\frac{V_d}{V}\left(\frac{2}{\beta}\right)^{\frac{d}{2}}\log\left(\frac{eV}{V_d}\left(\frac{\beta}{2}\right)^{\frac{d}{2}}\right)\right)\right). \tag{5}$$

*In particular, for fixed $\beta > 0$ and $d$, the bound grows with $M$ whenever $\beta > 0$ satisfies $|B(\mathbf{x})| \in (1, M]$ due to Proposition 2. Here, for any $\mathbf{x} \in \mathcal{X}$, a large $\beta$ leads to a small $|B(\mathbf{x})|$, and $|B(\mathbf{x})| > 1$ is the required condition for a novel local minima; a small $\beta$ leads to a large $|B(\mathbf{x})|$, and $|B(\mathbf{x})| \leq M$ stands for the case $B(\mathbf{x})$ at most covers the entire domain $\mathcal{X}$.*

The above result demonstrates that with the LSR energy, it is possible to exactly memorize all original patterns (with high probability) and still generate new patterns — what we term emergent memories (Definition 2). This behavior is surprising in the same way interpolating models in deep learning generalize unexpectedly well: both challenge the classical bias-variance intuition [15, 16]. While LSE-based models also produce novel memories, they typically do so at the expense of perfect recall of the original patterns. In Proposition 1 we show that LSE based DenseAMs do not have the global emergence property. This distinction highlights a key contribution of our work: *new memory creation need not come at the cost of perfect memorization.*

Next, we provide an exact order (i.e., upper and lower bounds) of the number of emergent memories under a grid design assumption.

**Proposition 3.** *If $\{\boldsymbol{\xi}_\mu\}_{\mu=1}^M$ form a grid over $\mathcal{X}$ of equal size with $\text{Vol}(\mathcal{X}) = V < \infty$, the number of emergent memories is of order $\Theta\left(\left(M^{1/d} - \lambda^{1/d} + 1\right)^d\right)$, where $\lambda = \Theta\left(MV^{-1}\left(8/\beta\right)^{\frac{d}{2}}\right)$ and for $\beta > 0$ such that $1 < \lambda \leq M$.*

Note that we showed an explicit form of the emergent memories $\mathbf{x}^* = \frac{1}{B(\mathbf{x}^*)}\sum_{\mu \in B(\mathbf{x}^*)}\boldsymbol{\xi}_\mu$, where $B(\mathbf{x}^*) = \left\{\mu : \|\mathbf{x}^* - \boldsymbol{\xi}_\mu\| < \sqrt{2/\beta}\right\} \subset \{1, \ldots, M\}$. equation 5 in Theorem 2 is stated under the uniform sampling regime, and Proposition 3 is stated under a fixed grid setting. In general, the number of emergent memories varies according to the specific geometry of the stored patterns $\{\boldsymbol{\xi}_\mu\}_{\mu=1}^M$, i.e., whether such a subset of $\{1, \ldots, M\}$ can be realized by a ball $\left\{\|\mathbf{x} - \boldsymbol{\xi}_\mu\| < \sqrt{2/\beta}\right\}$. This can grow much faster than a linear order of $M$, and is naively bounded by $2^M$.

## 4 Experiments

### 4.1 Quantifying the scaling of emergent memories

How many local minima do we see in practice as we: (a) vary the number of stored patterns, (b) change the dimensionality of those patterns, and (c) vary the inverse temperature $\beta$? We observe that, at critical values of $\beta$, we can create *orders of magnitude* more emergent memories than stored patterns. These results are shown in fig. 3 (left).

To quantify the number of local minima induced by the LSR energy, we uniformly sample $M$ patterns from the $d$-dimensional unit hypercube to serve as memories $\boldsymbol{\Xi}$. We enumerate all possible local minima of the LSR energy by computing the centroid $\bar{\boldsymbol{\xi}}_{\mathcal{K}} := |\mathcal{K}|^{-1}\sum_{\mu \in \mathcal{K}}\boldsymbol{\xi}_\mu$ for every possible subset of stored patterns $\mathcal{K} \subseteq [\![M]\!]$ (there are $2^M$ possible subsets if we allow for singleton sets). For each subset, we first check that its centroid is supported (i.e., that $E_\beta^{\text{LSR}}(\bar{\boldsymbol{\xi}}_{\mathcal{K}}; \boldsymbol{\Xi}) < \infty$ at $\epsilon = 0$), and then declare that $\bar{\boldsymbol{\xi}}_{\mathcal{K}}$ is a local minimum of the LSR energy if $\left\|\nabla E_\beta^{\text{LSR}}(\bar{\boldsymbol{\xi}}_{\mathcal{K}}; \boldsymbol{\Xi})\right\| < \delta$ for small $\delta > 0$. $\beta$ values are varied across the "interesting" regime between fully overlapping support regions (a single local minimum in the unit hypercube) to fully disjoint support regions around each memory. See experimental details in appendix D.2.

Certain values of $\beta$ yield particularly interesting behavior. For example, we observe that LSR can create orders of magnitude more emergent memories under ranges of $\beta$ where: (i) a majority ($>60\%$) of stored patterns are also recoverable, and (ii) around 20 percent of the unit hypercube is still

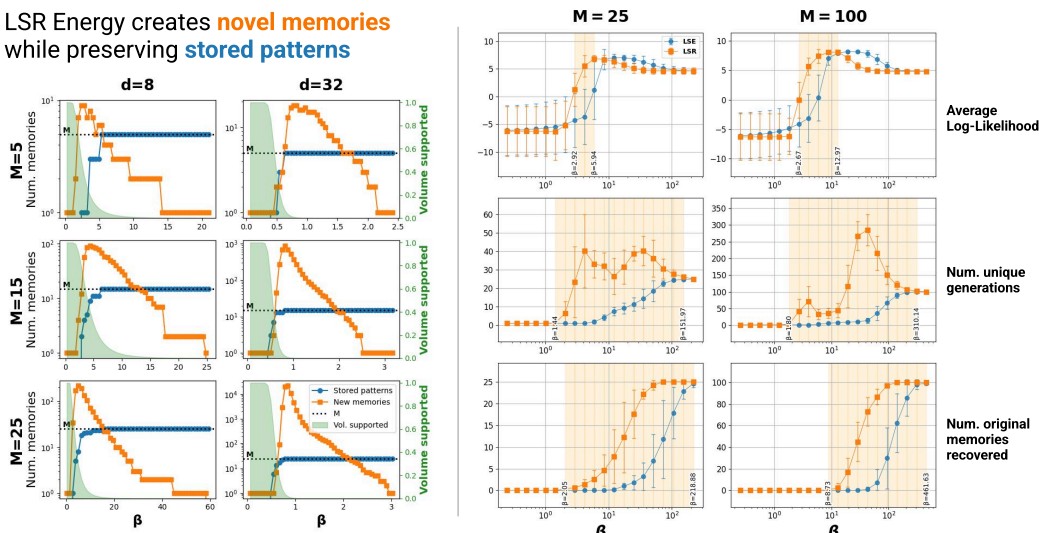

Figure 3: (Left) Analyzing local minima in LSR energy reveals a number of **novel memories** several orders of magnitude larger than $M$, the number of stored patterns, at critical values of $\beta$ (note that the y-axes are logscale). These emergent memories occur even while still preserving the **stored patterns** as memories. Smaller values of $\beta$ have a larger **region of support** on the unit hypercube. (Right) Given samples from some known true density function (in this case, a $k = 10$ mixture of 8-dim Gaussians with means drawn uniformly from the unit hypercube and $\sigma = 0.1$), memories from LSR energy have a log-likelihood comparable to, and occasionally slightly higher than, LSE under the true density function. Note that LSR achieves comparable log-likelihood while having more unique samples than LSE, even when both are seeded with the same $N = 500$ queries. Regions of $\beta$ where LSR outperforms LSE on a metric are specified by the orange regions. Error bars indicate the standard error across 5 different seeds for sampling stored patterns and initial queries.

supported. Note that in each experiment there are choices of $\beta$ such that the LSR energy does not exhibit *global emergence* (Definition 2, i.e., at low $\beta$ where novel memories are forming but not all stored patterns are yet retrievable). However, in these regions *local emergence* (Definition 3) of the novel memories still holds (see fig. 1 for intuition).

### 4.2 Generative quality of emergent memories

LSR memories are certainly more diverse than those of LSE, but do they represent more "meaningful" samples from a true, underlying density function $p(\mathbf{x})$ (as measured by their log-likelihood)? The experimental setup is as follows: Let $p(\mathbf{x})$ be a mixture of $k$ Gaussians whose means $\boldsymbol{\mu}_i \sim \mathcal{U}([0,1]^d)$ for $i \in [\![k]\!]$ are uniformly sampled from the $d$-dimensional unit hypercube with scalar ($\sigma = 0.1$) covariances such that $p(\mathbf{x}) = \frac{1}{k} \sum_{i=1}^{k} \mathcal{N}(\mathbf{x} \mid \boldsymbol{\mu}_i, \sigma^2 \mathbf{I}_d)$. We sample $M$ points $\{\boldsymbol{\xi}_1, \ldots, \boldsymbol{\xi}_M\}, \boldsymbol{\xi}_\mu \sim p(\mathbf{x})$ to serve as the stored patterns $\boldsymbol{\Xi}$ used to parameterize both the LSE and LSR energies from eq. (3). Define a thin *support boundary* induced by pattern $\boldsymbol{\xi}_\mu$ to be $\text{supp}[\boldsymbol{\xi}_\mu; \delta] = \{\mathbf{x} : 2\beta^{-1} - \delta \leq \|\mathbf{x} - \boldsymbol{\xi}_\mu\|^2 < 2\beta^{-1}\}$ for some small $\delta > 0$. Then, for initial points $\mathbf{x}_n^{(0)}, n \in [\![N]\!]$ sampled from the support boundary around each stored pattern,[2] LSE memories can be found using gradient descent

$$\mathbf{x}_n^{(t)} = \mathbf{x}_n^{(t-1)} - \alpha \nabla E_\beta^{\text{LSE}}(\mathbf{x}_n^{(t-1)}; \boldsymbol{\Xi}), \tag{6}$$

until convergence to a *memory* $\mathbf{x}_n^\star$. We use algorithm 1 to efficiently find the LSR memory corresponding to each initial point. Thus we have $N$ "samples" (memories) from both LSE and LSR on which we compare three metrics of interest in fig. 3 (right):

1. **Average Log-Likelihood**. Do LSR memories $\mathbf{x}_{\text{LSR}}^\star$ have higher $\log p(\mathbf{x}_{\text{LSR}}^\star)$ than LSE memories?
2. **Number of Unique Samples**. Does high $\log p(\mathbf{x}_{\text{LSR}}^\star)$ occur alongside many emergent memories?

---

[2] We use the same initial points to seed the dynamics of both $E^{\text{LSR}}$ and $E^{\text{LSE}}$. See appendix D.3 for details.

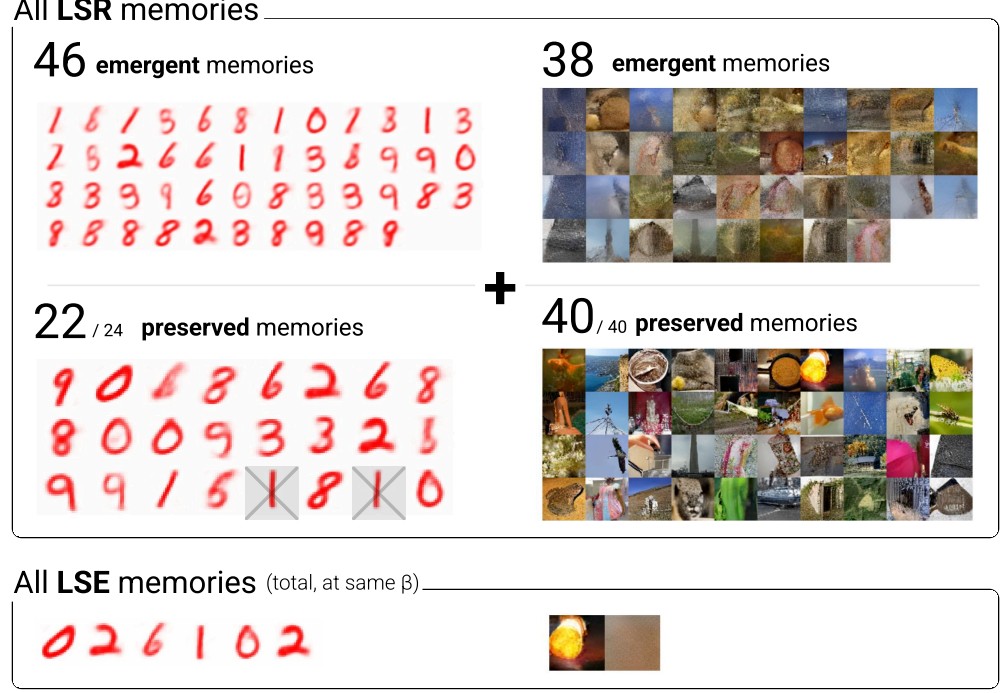

Figure 4: LSR's emergent memories appear as novel, creative generations when the energy is applied to a semantically meaningful latent space. **(Left)** 24 randomly-selected MNIST images are encoded into 10-dim VAE latents and stored into an LSR- and LSE-energy using a carefully chosen $\beta$ (see algorithm 2). Gray boxes indicate which stored patterns were not preserved at the chosen $\beta$. **(Right)** 40 TinyImagenet [27] images are encoded into 256-dim latents using a pretrained VAE [28] and stored into an LSR- and LSE-energy using a carefully chosen $\beta$. Note that in this TinyImagenet example the LSR energy is, by definition, *globally emergent* since all stored patterns are recoverable, while the MNIST example is not. See experiment details in appendix D.4.

3. **Number of Original Memories Recoverable**. Does high $\log p(\mathbf{x}_{\text{LSR}}^\star)$ occur when alongside high numbers of preserved memories? How does this trend compare with LSE memory performance?

The results tell a consistent story. Despite LSE energy being a more natural choice to model the underlying Mixture of Gaussians' density $p(\mathbf{x})$ (LSE has a Gaussian kernel that makes it ideal for the modeling task), LSR can match LSE in log-likelihood while simultaneously generating more diverse samples and preserving the stored patterns. See appendix D.3 for more experimental results and extended discussion.

## 4.3 Emergent memories in latent space

What do emergent memories look like when LSR is applied to real-world datasets? To study this behavior, we use a VAE to encode MNIST and TinyImagenet [27] images into latent vectors that serve as the stored patterns for LSR and LSE energies (see fig. 4). Using a carefully chosen $\beta$, we compute *all* memories (both preserved and novel) for each energy. The emergent memories of LSR in principle are simply the centroids of small subsets of the stored patterns, yet when decoded they appear as plausible and creative generations.

With the same $\beta$ value, LSR generates an order of magnitude more total memories than LSE. While this choice of $\beta$ is somewhat arbitrary and could be tuned separately for each energy, LSE would only ever be able to retrieve up to $M$ memories (where $M$ is the total number of stored patterns). See full experiment details in appendix D.4.

Emergent memories are mechanistically simple: they are simply the centroids of small subsets of the stored patterns. The semantic novelty of the emergent memories in Figure 4 occur because the latent space is structured to be semantically meaningful, where averaging two or more stored patterns

produces seemingly novel semantics. We conduct a similar experiment in appendix D.5 where we ablate the VAE to show what emergent memories look like in pixel space.

## 5   Discussion

Emergent memories are a powerful tool for creating novel samples, but the "meaningfulness" of these samples is a nuanced question that depends heavily on the specific application domain and task requirements. For example, in Figure 3 (right) we show that the LSR energy approximates an unknown p.d.f. better than LSE's energy while simultaneously generating diverse samples. This represents a desirable behavior of emergence, since high log-likelihood samples from the LSE energy are quite homogeneous, causing 500 initial queries to converge to the same ~10 memories. In this density estimation context, the emergent memories serve a clear functional purpose: they capture meaningful interpolations within the data distribution that improve generalization.

However, consider the novel memories from Tiny Imagenet in Figure 4. Though visually plausible, many of the emergent generations appear blurry and would be considered "undesirable" from the perspective of a high-fidelity image generation model. We discuss the limitations of emergent memories further in appendix A.

We note that there are potential parallels between emergent memories and "hallucinations" as observed in LLMs. We discuss the philosophical similarities between emergent memories and hallucinations in appendix B.1. It is also interesting to note that the LSR Energy presented in this work can have a feasible biological implementation using bipartite neurons. See further discussion in appendix B.2.

Finally, we reiterate that there are many choices for alternative kernels, and not just the Epanechnikov kernel. We discover that many kernels with compact support are capable of producing emergent memories and even manifolds, and the energy landscapes they produce can look quite different from each other. To this end, we show the "basin merging" behavior across different kernels in 1D in Figure 6 and we include an extended discussion on these other kernels in appendix C.

## 6   Conclusion

Our work introduces the LSR energy function which achieves the surprising combination of exact memorization of exponentially many patterns and the emergence of new, meaningful memories, thereby providing a powerful alternative to traditional AM formulations. The properties of the LSR energy define a novel class of *emergent memories* in Dense Associative Memory systems — a phenomenon not observed in any previous AM formulations. Unlike conventional models (e.g., LSE-based energies) where generalization (formation of local minima different from training data) typically comes at the cost of perfect memorization, LSR demonstrates that these objectives can coexist harmoniously in a single energy function. We additionally demonstrated that the diverse memories created by LSR achieve log-likelihood comparable to LSE when sampling from a true density function, while generating an order of magnitude more unique memories. Finally, we showed that when applied to latent representations of real-world image datasets, LSR's emergent memories represent plausible and creative generations.

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

# A    Limitations

To enable both emergence and exact memorization, the LSR energy studied in this work comes with certain limitations. Unlike the LSE energy whose gradient remains nonzero regardless of the query's distance from the stored patterns, the gradient of LSR energy vanishes exactly outside the support of the stored pattern. This makes gradient-based retrieval ineffective when the query lies far from any memorized pattern (though one can easily introduce a query-dependent temperature parameter $\beta(\mathbf{x})$ that dynamically adjusts to ensure the query lies within at least one basin of attraction). Alternatively, one can create a "hybrid" energy function that combines the LSE and LSR energies, taking advantage of LSE's non-zero gradient everywhere outside LSR's support around the stored patterns. Such temperature-tuned DenseAMs and hybrid energies are of independent interest and we leave their systematic study to future work.

Additionally, we want to emphasize that LSR can only create emergent memories precisely at the centroid of overlapping basins between stored patterns. This makes it possible to quickly and exactly retrieve memories (see Algorithm 3), but it also means that the "creativity" of this model is limited to a predictable subset of the convex hull of the stored patterns. The apparent novelty and creativity of Figure 4 is strongly aided by the semantic structure contained the VAEs' latent space.

# B    Extended Discussion

## B.1    On the relationship between emergent memories and hallucination

In everyday use of LLMs, we call it a *hallucination* when the model confidently produces an incorrect fact, especially when we know the model is capable of producing the correct fact in other settings. To formalize this idea into the context of emergent memories, we must imagine a setting where the model has an explicit energy function and a known set of stored "facts."

This is the setting of Associative Memory as studied in this work, where the outputs of our model are always energy minima (memories). A "hallucination" can occur when two or more stored facts interfere, leading the system to settle into an emergent minimum that looks meaningful (i.e., it lies near the "factual manifold" that generated our stored patterns) but does not correspond to any true, stored fact. Emergent memories are precisely these "interference minima" when they coexist with the original stored patterns. By definition, they are not stored facts, yet they can resemble meaningful combinations of them — making them the memory-system analog of hallucinations.

## B.2    On a biological implementation of emergent memories

Dense AMs permit a standard mapping onto "biological" networks by introducing auxiliary hidden neurons following the method of [8]. Thus, the feature neurons in both LSE and LSR models can be augmented with auxiliary hidden neurons, resulting in a model with only pair-wise neuron interactions and bipartite connectivity between feature and hidden layers. This augmented model is more biological compared to the model considered in the main paper. The proposed model defined by the energy eq. (3) and the corresponding update rule eq. (7) can be obtained by integrating out these auxiliary hidden neurons, which means LSR energy will still have the same emergent behavior. The augmented model will still have some un-biological aspects because of the weight symmetry between forward and backward projections, which is necessary to ensure that the network has a global energy function.

# C    On Kernels

We show different kernels that are typically used for KDE and their efficiency relative to the Epanechnikov kernel in fig. 5. See the explanation on optimal kernel density estimation in section 2 for more details.

Though our main submission focuses on the Epanechnikov kernel, we find that the phenomenon of emergence (Definitions 2 and 3) is not limited to the Epanechnikov kernel. We state our findings for other popular kernels below and graphically in Figure 6. In summary, all compact support kernels are capable of producing emergent memories *except* for the TriWeight kernel (meanwhile, the uniform kernel is impractical for associative memory). Details below:

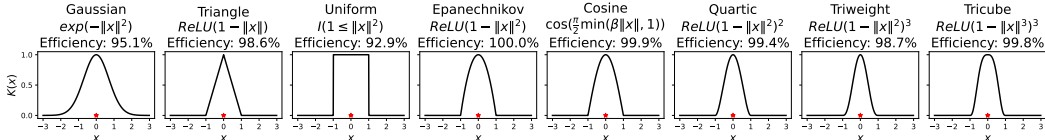

Figure 5: Different kernels used in KDE with their expression and KDE efficiency relative to the Epanechnikov kernel (*higher is better*, see text for details). The center of each kernel is marked with a red ⋆. To highlight the shape of the kernel, we have removed any scaling in the kernel expression. Note that all above kernels except Gaussian have finite support. The Epanechnikov kernel has the highest efficiency (100%). While the Gaussian kernel is extremely popular, and it is more efficient (95.1%) than the Uniform kernel (92.9%), there are various other kernels with better efficiency.

1. **Triangle kernel** — The triangle kernel exhibits very interesting emergence behavior. A perfectly *flat* energy manifold is formed where two or more basins overlap. If the energy of this manifold is lower than the energy of the stored patterns, the stored patterns "merge" and are no longer retrievable. If the energy of the manifold is higher, both original patterns are preserved and we observe a local *emergent manifold* of memory.
2. **Uniform kernel** — A uniform kernel produces an energy landscape that is impractical for associative memory because the gradient is zero everywhere except at the discontinuities. Emergent minima exist according to Definition 3 and have exclusively lower energy than the stored patterns.
3. **Triweight kernel** — We were not able to find emergence at any temperature with the triweight kernel, despite its compact support. Indeed, looking at the $\beta = 0.19$ row of Figure 6, we see that the transition of two basins merging results in a single, almost flat minimum. This phenomenon of compact support without emergence (or perhaps, emergence that is limited to an extremely narrow range of $\beta$ that we were not able to find) requires further investigation.
4. **Quartic kernel** — The quartic kernel produces smoother energy landscapes than the Epanechnikov does, and emergence appears within a very narrow range of $\beta$. Unlike the Epanechnikov kernel, overlapping basins do not guarantee emergent minima, and the emergent minimum is unlikely to have lower energy than the stored patterns.
5. **Tricube kernel** — The tricube kernel behaves like the Quartic kernel, but emergent memories are likely to have lower energy than the stored patterns. The Tricube kernel has an interesting property where local minima flatten right before basins merge. The resulting energy landscape is smoother than that of the Epanechnikov kernel.
6. **Cosine kernel** — The cosine kernel looks remarkably like the Epanechnikov kernel, and its emergence properties are almost identical to that of LSR. However, it appears that Cosine-emergent memories have slightly higher energy than their Epanechnikov counterpart.

# D    Experimental Details

## D.1    Reproducibility & Technical Resources

The codebase is published in a GitHub repository and contains necessary instructions to setup the environment and to recreate all results, down to the same seed used for training and sampling. Experiments use both PyTorch [29] and JAX [30]. Each experiment was run on a system with access to 8xA40 GPUs each with 40GB of memory. The log-likelihood experimental sweep in appendix D.3 took ∼12 hours using an optimized scheduler across 45 available CPU cores (memory turned out to be the bottle neck and the CPUs had access to much more RAM than the GPU). Searching for all novel minima at different $\beta$ and $M$ in appendix D.2 took ∼48 hours using an optimized scheduler across all 8 GPUs. Training the VAE in appendix D.4 took < 30 min on a single GPU; enumerating all local minima using the efficient algorithm 2 and algorithm 3 took < 15 min.

## D.2    Details: Quantifying Novel Minima

In this experiment we tested across a geometrically spaced range of $\beta \in [2d^{-1}, 2r_{\min}^{-2}]$, where $r_{\min} := \min_{\mu \neq \nu} \|\boldsymbol{\xi}_\mu - \boldsymbol{\xi}_\nu\|$ is the minimum pairwise distance between any two stored patterns in the current subset $\mathcal{K} \subseteq [\![M]\!]$. At the largest $\beta$, the support regions of the stored patterns are disjoint and the only memories are the $M$ stored patterns themselves; this configuration has a very small support region (shown as the shaded green curve in fig. 3, which is computed by monte carlo sampling 1e6

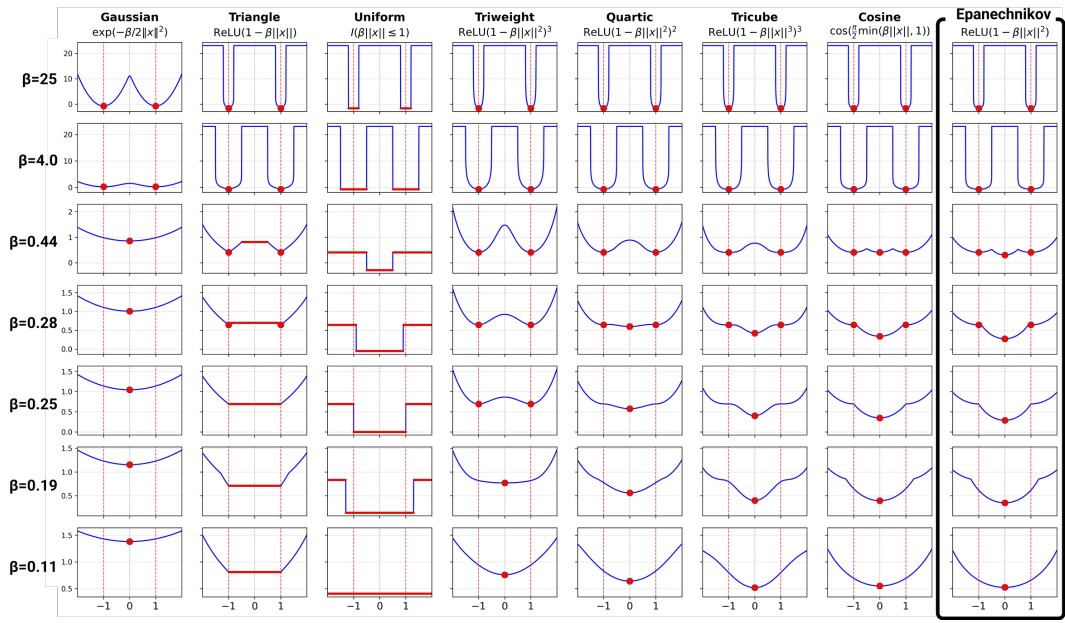

Figure 6: Comparing emergence across different choices of kernel in the DenseAM energy function. Emergent memories are highlighted in red, where manifolds are shown as a flat line and single points as larger dots. Interestingly, all compact kernels exhibit some form of emergence *except* the TriWeight kernel.

points on the unit hypercube and computing the fraction of energies that are finite at $\epsilon = 0$) as a fraction of the unit hypercube. At the smallest tested $\beta$, only a single energy minimum is induced at the centroid of all stored patterns with a region of support covering the whole unit hypercube. At the largest tested $\beta$, all original memories are recoverable and there are no spurious memories.

### D.3 Details: Generative quality of memories

#### D.3.1 Additional experiments

We ran the same experiment in section 4.2 under varying dimensions $d = [8, 16]$ and number of mixtures $k = [5, 10]$, averaging the results of each run across 5 different random seeds. The results for each of these experiments is shown below in figs. 7 and 8, where fig. 8 (left) is the same as reported in fig. 3 (right) in the main paper.

#### D.3.2 Aligning $\beta$'s across random seeds

We used the same setup for choosing an interesting range of $\beta$ as we did in appendix D.2. However, over random seeds the value for $r_{\min}$ can vary since it is dependent on the random choice of stored patterns. This makes it difficult to plot error bars over an individual $\beta$ across seeds. To fix this, we use the fact that each experiment has the same number of geometrically spaced $\beta$'s that start from the same $\beta_{\text{low}}$ and compute statistics averaged across the $\beta$'s that share an index. The $x$-axis then represents the $\beta$ value for each index averaged over seeds.

#### D.3.3 Determining the uniqueness of minima

To sample the LSE minima, 500 initial points are uniformly sampled from the support boundary around each memory and gradient descent eq. (6) is performed for 13000 steps at a cosine-decayed learning rate $\alpha$ from $0.01 \rightarrow 0.0001$. However, even after this descent process, there are variations in the retrieved memories due to discrete step size $\alpha$ and floating point precision requiring us to be careful when deciding if two samples are distinct. Generally, memory retrieval is said to converge when $\|\nabla_{\mathbf{x}} E(\mathbf{x})\| < \epsilon$ for some small $\epsilon > 0$, or when the number of iterations $T$ exceeds some threshold at small $\alpha$. Because we know the $\beta$ of the LSE energy, by the properties of the Gaussian kernel we know that two basins merge when their means are within two standard deviations of each

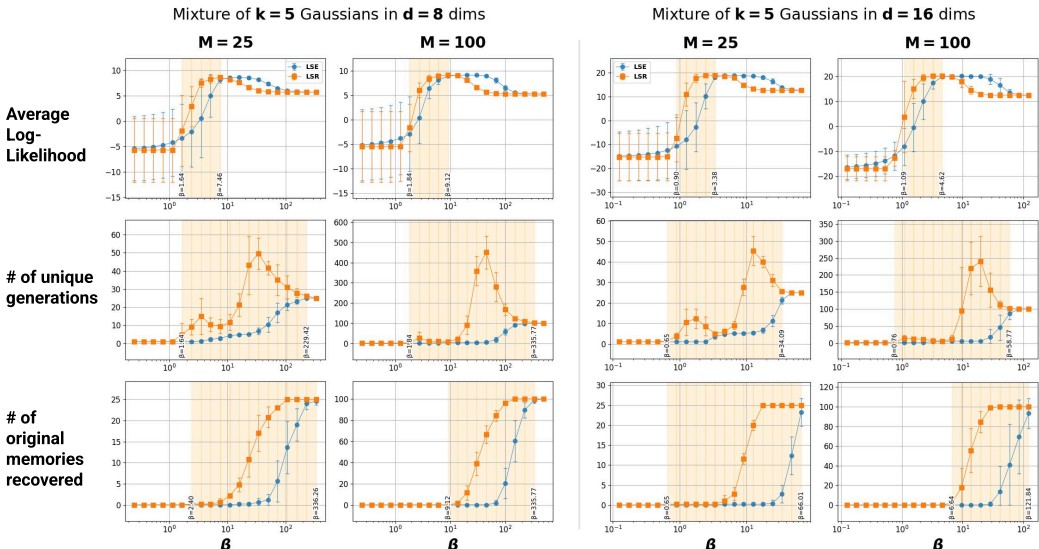

Figure 7: Comparing $d = 8$ and $d = 16$ for $k = 5$ mixture of gaussians at number of stored patterns $M = 10$ and $M = 100$. Error bars are computed by averaging the results of 5 different random seeds. Regions of $\beta$ where LSR outperforms LSE on a particular metric (on average) are shaded orange.

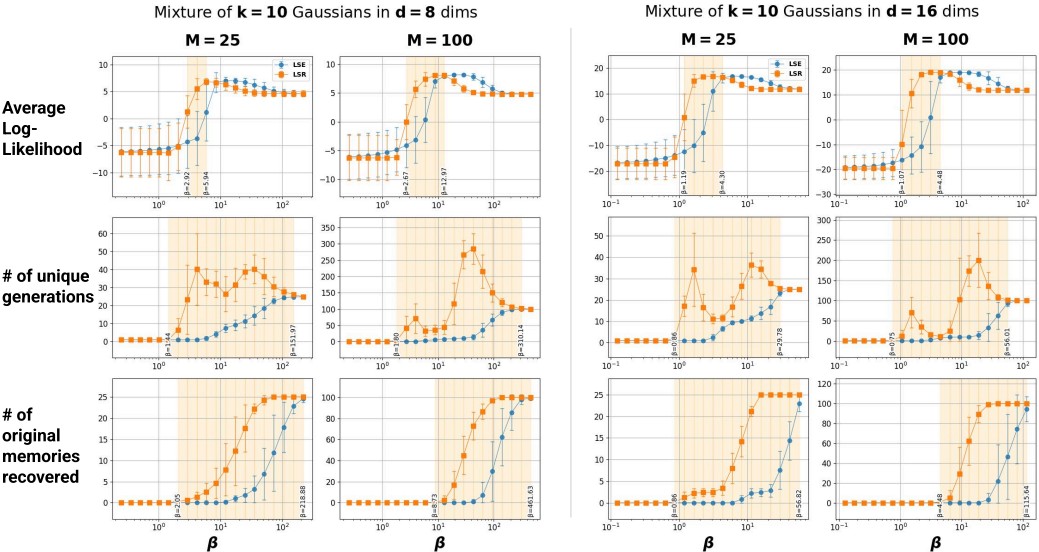

Figure 8: Comparing $d = 8$ and $d = 16$ for $k = 10$ mixture of gaussians at number of stored patterns $M = 10$ and $M = 100$. Error bars are computed by averaging the results of 5 different random seeds. Regions of $\beta$ where LSR outperforms LSE on a particular metric (on average) are shaded orange.

other. Thus, we can say that two distinct samples are generated by the same memory if they are within $\frac{2}{\sqrt{\beta}}$ of each other.

When counting the uniqueness of the samples from the LSR energy, we perform the following trick to exactly compute the fixed points of the dynamics. We first compute our "best guess" for the fixed point by performing standard gradient descent according to eq. (6) for $T$ steps, at which point $\mathbf{z} := \mathbf{x}^{(T)}$ is close (but not exactly equal) to the fixed point $\mathbf{x}^\star$. We then pass $\mathbf{z}$ to algorithm 1 to compute the fixed point exactly. With a good initial guess $\mathbf{z}$, algorithm 1 converges after a single iteration.

**Algorithm 1:** Fixed Point Computation for the LSR Memory Retrieval

**Input:** Initial guess $\mathbf{z}$, stored patterns $\{\boldsymbol{\xi}_\mu\}_{\mu=1}^{M}$, inverse temperature $\beta$
**Output:** Fixed point $\mathbf{z}^\star$
Initialize previous point $\mathbf{z}_{\text{prev}} \leftarrow \mathbf{z} + \infty$
**while** $\mathbf{z}_{prev} \neq \mathbf{z}$ **do**
    $\mathbf{z}_{\text{prev}} \leftarrow \mathbf{z}$
    Compute supports near $\mathbf{z}$ $S(\mathbf{z}) \leftarrow \{\boldsymbol{\xi}_\mu : \|\mathbf{z} - \boldsymbol{\xi}_\mu\| \leq \sqrt{\frac{2}{\beta}}\}$
    Update to mean of support centroids $\mathbf{z} \leftarrow \frac{1}{|S(\mathbf{z})|} \sum_{\boldsymbol{\xi}_\mu \in S(\mathbf{z})} \boldsymbol{\xi}_\mu$
**end**
**return** $\mathbf{z}$

Finally, we choose to sample points near the support boundary of each stored pattern because this maximizes the probability that we will end up in a spurious minimum. The size of spurious basins in high dimension can be very small, and the probability of landing in them decreases rapidly with increasing $\beta$ (see the region of support plot in fig. 3).

### D.4 Details: Qualitative reconstructions

The experiments in fig. 4 are conducted in two steps. First, we **design an energy** for each dataset. Then, we **discover all memories** for each system.

**Designing the energy** The DenseAM energies studied in this work are described by a matrix of stored patterns $\Xi$ and an inverse temperature hparam $\beta$.

*Choosing $\Xi$.* For MNIST, $\Xi$ is obtained as follows: 24 random images from the MNIST training set are normalized to be $[0, 1]$, rasterized into a 784-dim vector, and projected into a 10-dim latent space of a $\beta$-VAE trained according to the methods laid out below. These latents become our stored patterns $\Xi_{\text{MNIST}} \in [0, 1]^{24 \times 768}$ that we use to parameterize both the LSR energy and the LSE energy. The procedure for TinyImagenet [27] is similar. We randomly select 40 images from the dataset, each of shape `(C,H,W)=(3,64,64)`. These samples are passed through a small pretrained VAE called TAESD [28] to produce latents that are of shape `(4,8,8)` which are then rasterized into vectors of shape (256,). Thus, our stored pattern matrix for TinyImagenet is $\Xi_{\text{TinyImgnet}} \in \mathbb{R}^{40 \times 256}$.

*Choosing $\beta$* Tuning $\beta$ for the LSR energy is challenging. When $\beta$ is too small, each stored pattern interacts with all other stored patterns to induce one minimum at the centroid and several minima far from the data distribution; when $\beta$ is too large, each stored pattern will interact with no other stored patterns and will induce only a single minimum at itself. Neither of these regimes are interesting. For large ranges of $\beta$ between these limits, the combinatorial search space of possible memories is computationally prohibitive. For this reason, we use a binary search algorithm to choose a $\beta$ that, on average, causes each stored pattern to interact with approximately 4 other stored patterns (in the 10-dim MNIST case) and 2 other stored patterns in the 256-dim TinyImagenet case. This $\beta$ encourages the LSR energy to exhibit emergence where it is computationally feasible to enumerate all memories, and we use it for both the LSR and LSE energy experiments. In pseudocode, the search algorithm is given by algorithm 2

*Training the $\beta$-VAE for MNIST.* MNIST images are encoded by a $\beta$-VAE [31, 32] with a latent dimension of 10. The VAE takes as input the 784-dim rasterized MNIST images. The VAE's encoder and decoder are two layer MLPs configured with LeakyReLU and BatchNorm activations, with a hidden dimension size of 512. Training proceeded for 50 epochs using a learning rate of 1e-3, the Adam optimizer, and a minibatch size of 128. The $\beta$ of the $\beta$-VAE (distinct from the inverse temperature $\beta$ used by the LSR energy) is set to 4.

**Discovering all memories** Once we have chosen a $\beta$ that results in a feasible number of basin interactions $K$ (where the combinatorial search space is tractable), all memories for LSR are discovered by filtering the set of "memory candidates" — the set of centroids formed by at overlapping basins around each stored patterns — to those whose energy gradient is zero. This method is explicitly described in algorithm 3, which iterates through each stored patterns $\boldsymbol{\xi}_\mu$ and only searches for emergent memories formed by near-enough stored patterns.

**Algorithm 2:** Binary Search over $\beta$ to achieve desired memory interactions

---

**Input:** Desired avg. interactions per pattern $K$, stored patterns $\{\boldsymbol{\xi}_\mu\}_{\mu=1}^M$, max iterations $n_{\max}$
**Output:** Optimal $\beta^*$ achieving number of basin interactions $K$
**Compute**
    |   Distance matrix $D_{\mu\nu} \leftarrow \{\|\boldsymbol{\xi}_\mu - \boldsymbol{\xi}_\nu\|\}$
    |   Binary bounds $(r_{\min}, r_{\max}) \leftarrow (0.5\min(D), 4\max(D))$
    |   Initial basin radius $r \leftarrow \mathrm{mean}(r_{\min}, r_{\max})$
**end**
$n_{\mathrm{iter}} \leftarrow 0$
**repeat**
    |   Compute avg. number of interacting basins per memory
    |     $K' \leftarrow \mathrm{mean}_\mu \left( \sum_{\nu=1}^M \mathbb{1}[D_{\mu\nu} \leq 2r] \right)$
    |   Update binary search conditions
    |     $r_{\min} \leftarrow r$ if $K' < K$
    |     $r_{\max} \leftarrow r$ if $K' > K$
    |     $r \leftarrow \mathrm{mean}(r_{\min}, r_{\max})$
    |   $n_{\mathrm{iter}} \leftarrow n_{\mathrm{iter}} + 1$
**until** $K' = K$ or $n_{iter} \geq n_{max}$
$\beta^* \leftarrow 2/r^2$
**return** $\beta^*$

---

**Algorithm 3:** Discover local minima of the LSR energy at a specific $\beta$.

---

**Input:** Stored patterns $\{\boldsymbol{\xi}_\mu\}_{\mu=1}^M$, inverse temperature $\beta$, gradient norm threshold $\delta$ near 0
**Output:** Set of LSR memories $\mathcal{X}^*$.
**Compute**
    |   Distance matrix $D_{\mu\nu} \leftarrow \{\|\boldsymbol{\xi}_\mu - \boldsymbol{\xi}_\nu\|\}$
    |   Basin radius $r \leftarrow \sqrt{2/\beta}$
**end**
Initialize set of local minima $\mathcal{X}^* \leftarrow \emptyset$
**for** $\mu \in [\![M]\!]$ **do**
    |   Compute set of interacting neighbors $\mathbf{X}_\mu \leftarrow \{\boldsymbol{\xi}_\nu : D_{\mu\nu} \leq 2r\}$
    |   Compute the set of all non-empty subsets $\mathcal{C} \leftarrow \{S \subseteq \mathbf{X}_\mu : \mathrm{size}(S) > 0\}$
    |   **for** $S \in \mathcal{C}$ **do**
    |     |   Compute centroid of neighbors $\bar{\mathbf{x}}_S \leftarrow \mathrm{mean}(S)$
    |     |   Compute $\bar{\mathbf{x}}_S$ neighbors $T_S \leftarrow \{\boldsymbol{\xi}_\nu : \|\boldsymbol{\xi}_\nu - \bar{\mathbf{x}}_S\| \leq r\}$
    |     |   **if** $\|\nabla E_{\mathrm{LSR}}(\bar{\mathbf{x}}_S)\| < \delta$ **&** $E_{\mathrm{LSR}}(\bar{\mathbf{x}}_S) < \infty$ **then**
    |     |     |   Update set of local minima $\mathcal{X}^* \leftarrow \mathcal{X}^* \cup \{\bar{\mathbf{x}}_S\}$
    |     |   **end**
    |   **end**
**end**
**return** $\mathcal{X}^*$

---

All LSE memories are discovered via gradient descent. We initialize queries $\mathbf{x}_\mu = \boldsymbol{\xi}_\mu$ and perform gradient descent according to eq. (6) until convergence (for this experiment, we iterated for 20k steps at small step-size $\alpha = 0.002$). The retrieved fixed points are the complete set of LSE memories.

### D.5 Additional experiment: Pixel-space emergence

When the stored patterns live in a semantically structured latent space, as is done in fig. 4, the energy landscape inherits the structure such that centroids of subsets of stored patterns appear semantically novel. However, the latent space visually obscures the mechanistic simplicity of emergent memories. Thus, we repeat the experiment in pixel space to reinforce how emergent memories work.

We store 8 randomly selected MNIST images as rasterized pixels (normalized between 0 and 1) into the LSR energy. The resulting emergent minima and stored patterns are shown in Figure 9, where the $\beta$ is chosen to balance the number of emergent minima and the retrievability of the stored patterns (*global emergence* Definition 2).

### D.6 Additional experiment: Scaling number of stored patterns

The LSR energy function is simple and its properties hold across any scale of stored patterns. To show this, we store all 60,000 MNIST training images into the LSR energy and select a $\beta$ for which $\sim$50% of the stored patterns are still retrievable. We select a "seed" image at random and randomly select 15 images from all images whose basins interact with the seed image at the chosen $\beta$. This example is shown in Figure 10.

## E Proofs

### E.1 Proof of Theorem 1

For any $\mathbf{x} \in \mathcal{X}$, let $B(\mathbf{x}) = \{\mu \in [\![M]\!] : \|\mathbf{x} - \boldsymbol{\xi}_\mu\|^2 \leq 2/\beta\}$. Then the gradient of the LSR energy in eq. (3) is given by

$$\nabla_{\mathbf{x}} E^{\text{LSR}}(\mathbf{x}) = \sum_{\mu=1}^{M} \frac{(\mathbf{x} - \boldsymbol{\xi}_\mu)\, \mathbb{1}\left[\|\mathbf{x} - \boldsymbol{\xi}_\mu\|^2 \leq \frac{2}{\beta}\right]}{\epsilon + \left[\sum_{\nu=1}^{M} \text{ReLU}\left(1 - \frac{\beta}{2}\|\mathbf{x} - \boldsymbol{\xi}_\nu\|^2\right)\right]} \tag{7}$$

$$= \frac{\sum_{\mu \in B(\mathbf{x})}(\mathbf{x} - \boldsymbol{\xi}_\mu)}{\epsilon + \left[\sum_{\nu \in B(\mathbf{x})} \text{ReLU}\left(1 - \frac{\beta}{2}\|\mathbf{x} - \boldsymbol{\xi}_\nu\|^2\right)\right]} \tag{8}$$

With $\beta = 2/(r-\Delta)^2$, $B(\mathbf{x}) = \{\mu \in [\![M]\!] : \|\mathbf{x} - \boldsymbol{\xi}_\mu\| \leq (r-\Delta)\}$. For any $\mathbf{x} \in S_\mu(\Delta)$, $B(\mathbf{x}) = \{\mu\}$. Thus the LSR energy gradient simplifies to

$$\forall \mathbf{x} \in S_\mu(\Delta), \quad \nabla_{\mathbf{x}} E^{\text{LSR}}(\mathbf{x}) = \frac{(\mathbf{x} - \boldsymbol{\xi}_\mu)}{\epsilon + \text{ReLU}\left(1 - \frac{\beta}{2}\|\mathbf{x} - \boldsymbol{\xi}_\mu\|^2\right)}, \tag{9}$$

which is exactly zero at $\mathbf{x} = \boldsymbol{\xi}_\mu$, thus giving us the retrieval of the $\mu^{\text{th}}$ memory via energy gradient descent.

Furthermore, again for $\mathbf{x} \in S_\mu(\Delta)$ with a energy gradient descent learning rate set to $\eta \leftarrow \epsilon + \text{ReLU}\left(1 - \beta/2\|\mathbf{x} - \boldsymbol{\xi}_\mu\|^2\right)$, the update is exactly $\eta \nabla_{\mathbf{x}} E^{\text{LSR}}(\mathbf{x}) = (\mathbf{x} - \boldsymbol{\xi}_\mu)$. Thus a single step gradient descent update to $\mathbf{x}$ with $\mathbf{x} - \eta \nabla_{\mathbf{x}} E^{\text{LSR}}(\mathbf{x}) = \mathbf{x} - (\mathbf{x} - \boldsymbol{\xi}_\mu) = \boldsymbol{\xi}_\mu$ results in the retrieval of the $\mu^{\text{th}}$ memory.

### E.2 Proof of Proposition 2

*Proof.* (Proposition 2.)

For any $\mathbf{x} \in \mathcal{X}$, given the definition of $B(\mathbf{x})$, recall that the gradient of the LSR energy is given in eq. (7). This gradient is zero when

$$\mathbf{x} = \frac{1}{|B(\mathbf{x})|} \sum_{\mu \in B(\mathbf{x})} \boldsymbol{\xi}_\mu,$$

Figure 9: **Emergent memories** are centroids of subsets of stored patterns, shown clearly when 8 stored images are visualized in pixel space alongside their induced emergent memories. Stored patterns (bottom, indexed **A**-**H**) merge to form emergent memories (top, labeled by the stored patterns that merged to form the emergent memory). $\beta$ is chosen such that the number of emergent memories is approximately the same as the number of stored patterns.

**Seed image**  **Nearby emergent memories**

Figure 10: Sampling emergent memories near a **seed image** when all 60k MNIST training images are stored into the LSR energy. Left: Random seed image, which is a preserved memory. Right: 16 randomly sampled emergent memories formed by the seed image's interactions with other stored patterns (at $\beta = 0.11$). Because the seed image interacts with the basins of $\sim$7.3k other stored patterns, these emergent memories represent a *tiny sample* of the total emergent memories near the seed image.

the geometric mean of the memories corresponding to the set $B(\mathbf{x})$. Moreover, by standard algebraic computations, we have

$$\nabla_{\mathbf{x}}^2 E_{LSR}(\mathbf{x}^*) = \frac{|B(\mathbf{x}^*)|}{\epsilon + \left[\sum_{\nu=1}^{M} \mathrm{ReLU}\left(1 - \frac{\beta}{2}\|\mathbf{x}^* - \boldsymbol{\xi}_\nu\|^2\right)\right]} I_d \succ 0.$$

$\square$

### E.3  Proof of Theorem 2

Note that once we prove properties (a) and (b), the $\varepsilon$-global emergence in Definition 2 follows immediately.

**We first focus on (a).** Note that for any fixed $\mathbf{x} \in \mathcal{X}$ and $\varepsilon > 0$, $\mathrm{vol}(B(\mathbf{x}, \varepsilon) \cap \mathcal{X})$ is at most $\mathrm{vol}(B(\mathbf{x}, \varepsilon)) = V_d \varepsilon^d$, where $V_d$ is the volume of the unit ball in $\mathbb{R}^d$. Thus, we obtain: for any pair $(\boldsymbol{\xi}_\mu, \boldsymbol{\xi}_\nu)$ with $\mu \neq \nu$, we have:

$$\Pr\left[\|\boldsymbol{\xi}_\mu - \boldsymbol{\xi}_\nu\| < \varepsilon\right] \leq \int_{\mathcal{X}} \frac{\mathrm{vol}(B(x, \varepsilon) \cap \mathcal{X})}{\mathrm{vol}(\mathcal{X})} dx \leq V_d V^{-1} \varepsilon^d.$$

Then, we have:

$$\Pr\left[\min_{1 \leq \mu \neq \nu \leq M} \|\boldsymbol{\xi}_\mu - \boldsymbol{\xi}_\nu\| < \varepsilon\right] \leq M^2 V_d V^{-1} \varepsilon^d. \tag{10}$$

Now, we let $\varepsilon = (V_d/V)^{-1/d} e^{-2\alpha}$ for a positive $\alpha$. Thus, with probability at least $1 - M^2 e^{-2\alpha d}$, the minimum pairwise Euclidean distance $r = \min_{1 \leq i \neq j \leq M} \|\boldsymbol{\xi}_\mu - \boldsymbol{\xi}_\nu\| \geq (V_d/V)^{-1/d} e^{-2\alpha}$. Then, with

$$0 < \beta = \frac{2}{(r - \Delta)^2} \leq \frac{2}{((V_d/V)^{-1/d} e^{-2\alpha} - \Delta)^2},$$

we are able to retrieve any memory $\boldsymbol{\xi}_\mu$ with an $\mathbf{x} \in S_\mu(\Delta)$, for $\Delta \in (0, (V_d/V)^{-1/d} e^{-2\alpha})$. Moreover, the size of $M$ is given by the success probability:

$$\delta = 1 - M^2 e^{-2\alpha d}.$$

This gives

$$M = \Theta\left(\sqrt{1 - \delta} e^{\alpha d}\right).$$

**Next, we focus on (b).** Recall the gradient is given by

$$\nabla_{\mathbf{x}} E_{\mathrm{LSR}}(\mathbf{x}) = \frac{\sum_{\mu \in B(\mathbf{x})} (\mathbf{x} - \boldsymbol{\xi}_\mu)}{\epsilon + \left[\sum_{\mu \in B(\mathbf{x})} \mathrm{ReLU}\left(1 - \frac{\beta}{2}\|\mathbf{x} - \boldsymbol{\xi}_\mu\|^2\right)\right]},$$

where it is determined by the set

$$B(\mathbf{x}) = \left\{ \mu : \|\mathbf{x} - \boldsymbol{\xi}_\mu\| < \sqrt{\frac{2}{\beta}} \right\}.$$

Let $S_{\mathbf{x}^*}(\Delta) = \{\mathbf{x} : \|\mathbf{x} - \mathbf{x}^*\| \leq \Delta\}$ be a basin around an emergent memory $\mathbf{x}^*$. Note that by Cauchy–Schwarz inequality, the change within the basin of the squared distance to any $\boldsymbol{\xi}_\mu$ is at most

$$|\|\mathbf{x} - \boldsymbol{\xi}_\mu\|^2 - \|\mathbf{x}^* - \boldsymbol{\xi}_\mu\|^2| \leq 2\|\mathbf{x}^* - \boldsymbol{\xi}_\mu\|\Delta + \Delta^2.$$

We now for activated memories $\mu \in B(\mathbf{x}^*)$ let

$$\delta_{\min}(\mathbf{x}^*) = \min_{\mu \in B(\mathbf{x}^*)} \left( \frac{2}{\beta} - \|\mathbf{x}^* - \boldsymbol{\xi}_\mu\|^2 \right) > 0.$$

This is the margin to the boundary of activation. Moreover, for inactivated memories $\nu \notin B(\mathbf{x}^*)$, let

$$\gamma_{\min}(\mathbf{x}^*) = \min_{\nu \notin B(\mathbf{x}^*)} \left( \|\mathbf{x}^* - \boldsymbol{\xi}_\nu\|^2 - \frac{2}{\beta} \right) > 0.$$

This is the margin to being activated. Given $\{\boldsymbol{\xi}_\mu\}_{\mu=1}^M$ has a density, the probability of either of them being exact zero is zero.

Finally, we determine the radius $\Delta$. To ensure that for any $\mathbf{x} \in S_{\mathbf{x}^*}(\Delta)$, $\nabla_{\mathbf{x}}E_{\mathrm{LSR}}(\mathbf{x}) = \nabla_{\mathbf{x}}E_{\mathrm{LSR}}(\mathbf{x}^*)$, it suffices to make sure $B(\mathbf{x}) = B(\mathbf{x}^*)$ for any $\mathbf{x} \in S_{\mathbf{x}^*}(\Delta)$. To this end, we can pick $\Delta$ such that

$$2D_{\max}(\mathbf{x}^*)\Delta + \Delta^2 < \min\{\delta_{\min}(\mathbf{x}^*), \gamma_{\min}(\mathbf{x}^*)\}, \text{ where } D_{\max}(\mathbf{x}^*) = \max_\mu \|\mathbf{x}^* - \boldsymbol{\xi}_\mu\|.$$

This gives

$$\Delta < \sqrt{D_{\max}^2(\mathbf{x}^*) + \min\{\delta_{\min}(\mathbf{x}^*), \gamma_{\min}(\mathbf{x}^*)\}} - D_{\max}(\mathbf{x}^*) =: r^*.$$

**To prove the final claim**, without loss of generality, we consider the case when $\mathcal{X} = [0,1]^d$ with $V = 1$. Since $\{\boldsymbol{\xi}_\mu\}_{\mu=1}^M$ are i.i.d. uniform, $\mathbb{1}\left[\boldsymbol{\xi}_\mu \in \left\{\|\mathbf{x} - \boldsymbol{\xi}_\mu\| \leq \sqrt{\frac{2}{\beta}}\right\}\right]$ are i.i.d. indicators over $\mu = 1, \ldots, M$, by the concentration of sub-Gaussian random variables [33, Proposition 2.5], we have: with probability at least $1 - M^{-1}$,

$$\lambda = |B(\mathbf{x})|$$

$$\in \left[ MV_d \left(\frac{2}{\beta}\right)^{\frac{d}{2}} - 5\sqrt{MV_d \left(\frac{2}{\beta}\right)^{\frac{d}{2}} \log M}, \ MV_d \left(\frac{2}{\beta}\right)^{\frac{d}{2}} + 5\sqrt{MV_d \left(\frac{2}{\beta}\right)^{\frac{d}{2}} \log M} \right]. \quad (11)$$

This implies $\lambda = \Theta\left(MV_d \left(\frac{2}{\beta}\right)^{\frac{d}{2}}\right)$. Note that according to Proposition 2, for each $|B(\mathbf{x})| \in (1, M]$, there is a corresponding new emergent memory as the average of all stored patterns over $\mu \in B(\mathbf{x})$. Then, the number of new emergent memories is bounded as

$$Q_{\beta,\mathrm{LSR}} \leq \binom{M}{\lambda}, \text{ where } \lambda = \Theta\left(MV_d \left(\frac{2}{\beta}\right)^{\frac{d}{2}}\right).$$

By Stirling's formula, we have:

$$Q_{\beta,\mathrm{LSR}} \leq \left(\frac{eM}{\lambda}\right)^\lambda.$$

Thus, it holds that

$$\log Q_{\beta,\mathrm{LSR}} \leq \lambda \log\left(\frac{eM}{\lambda}\right) = O\left(MV_d \left(\frac{2}{\beta}\right)^{\frac{d}{2}} \log\left(\frac{e}{V_d}\left(\frac{\beta}{2}\right)^{\frac{d}{2}}\right)\right).$$

This implies

$$Q_{\beta,\mathrm{LSR}} = O\left(\exp\left(MV_d \left(\frac{2}{\beta}\right)^{\frac{d}{2}} \log\left(\frac{e}{V_d}\left(\frac{\beta}{2}\right)^{\frac{d}{2}}\right)\right)\right).$$

### E.4 Proof of Proposition 3

Without loss of generality, we consider the case when $\mathcal{X} = [0,1]^d$ with $V = 1$. Now, we divide $[0,1]^d$ into an equally spaced grid of size $M$, i.e., we divide each dimension $[0,1]$ into $[mM^{-1/d}, (m+1)M^{-1/d}]$ for $m = 0, \ldots, M-1$. Under our grid setting, $\{\boldsymbol{\xi}_\mu\}_{\mu=1}^M$ lie in such a grid of equal size.

Now, note that

$$B(\mathbf{x}) = \left\{ \mu : \|\mathbf{x} - \boldsymbol{\xi}_\mu\| < \sqrt{\frac{2}{\beta}} \right\}.$$

Suppose the ball in definition of $B(\mathbf{x})$ contains $\lambda$ of those equally spaced stored patterns such that $\lambda = |B(\mathbf{x})| = \lambda \in (1, M]$. We want to put a hypercube in the grid such that it contains exact $\lambda$ out of $M$ equally spaced stored patterns in the grid. Here, for simplicity, it is up to a constant depending on $d$ if we think of the ball as a hypercube in $\mathbb{R}^d$ containing the points in the grid in this case. Hence, in each dimension $1 \le i \le d$, we put an interval containing of order $\lambda^{1/d}$ points in the grid inside the entire interval containing of order $M^{1/d}$ points in the grid. It thus gives the number of choice for each dimension $1 \le i \le d$ as:

$$M^{1/d} - \lambda^{1/d} - 1$$

Hence, counting all choices over $d$ dimensions, we obtain the number of such $B(\mathbf{x})$ as

$$\left( M^{1/d} - \lambda^{1/d} - 1 \right)^d.$$

Moreover, due to the equal spacing of the grid, we have:

$$\lambda = \Theta\left( M \left( \frac{8}{\beta} \right)^{\frac{d}{2}} \right). \tag{12}$$

Hence, combining all bounds above, we obtain: the number of emergent memories under uniform sampling is of order

$$\Theta\left( \left( M^{1/d} - \lambda^{1/d} + 1 \right)^d \right),$$

where $\lambda$ satisfies equation 12 and $1 < \lambda \le M$.

### E.5 Proof of Proposition 1

Recall the LSE energy:

$$E^{\mathrm{LSE}}(\mathbf{x}) = -\frac{1}{\beta} \log \sum_{\mu=1}^M e^{-\frac{\beta}{2} \|\mathbf{x} - \boldsymbol{\xi}_\mu\|^2}.$$

Thus, the gradient is

$$\nabla_{\mathbf{x}} E^{\mathrm{LSE}}(\mathbf{x}) = \mathbf{x} - \sum_{\mu=1}^M p_\mu(\mathbf{x}) \boldsymbol{\xi}_\mu, \ p_\mu(\mathbf{x}) := \frac{e^{-\frac{\beta}{2} \|\mathbf{x} - \boldsymbol{\xi}_\mu\|}}{\sum_{\mu=1}^M e^{-\frac{\beta}{2} \|\mathbf{x} - \boldsymbol{\xi}_\mu\|}}.$$

Suppose $\boldsymbol{\xi}_\nu$ for some $\nu$ is the zero of this gradient, it yields that

$$(1 - p_\nu(\boldsymbol{\xi}_\nu)) \boldsymbol{\xi}_\nu = \sum_{\mu \neq \nu} p_\mu(\boldsymbol{\xi}_\mu) \boldsymbol{\xi}_\mu.$$

This is a contradiction of the assumption that $\{\boldsymbol{\xi}_\mu\}_{\mu=1}^M$ are linearly independent.

