# OpenReview forum: "Dense Associative Memory with Epanechnikov Energy"
_NeurIPS.cc/2025/Conference — NeurIPS 2025 spotlight_

### Official Review · Reviewer_4hP4 · 2025-06-25

**Clarity:** 4
**Significance:** 4
**Originality:** 4
**Rating:** 6
**Confidence:** 3

**Summary:**

This paper proposes a new energy function for Dense Associative Memory (DenseAM) models, called the Log-Sum-ReLU (LSR) energy, inspired by kernel density estimation (KDE) and particularly the Epanechnikov kernel. Classical and modern Hopfield networks rely on energy functions like log-sum-exp (LSE) to enable memory retrieval with high capacity. However, such models often trade off memorization for generalization. The authors show that LSR energy defies this trade-off: it enables exact retrieval of all stored memories and also generates emergent local minima—interpretable as novel “creative” memories—without compromising recall.

Theoretically, the authors demonstrate that LSR energy preserves exponential capacity and enables global emergence, where novel attractors appear even while all original patterns are retrievable. This contrasts with LSE, which lacks this capability under any fixed temperature $\beta$. The LSR energy is derived from a shifted ReLU separation function, corresponding to the optimal KDE kernel under certain assumptions.

Empirically, the paper validates these findings in toy datasets (uniform points in a hypercube), synthetic mixture-of-Gaussian tasks, and real-world datasets (MNIST and TinyImageNet), with memory retrieval occurring in VAE latent space. Experiments show that LSR generates more unique, semantically plausible memories than LSE, sometimes with better log-likelihood under the ground-truth density.

**Questions:**

I feel this paper is already above the acceptance threshold. However, I see the following potential opportunities that may promote this paper to a spotlight/oral one if the authors could include them:

- Provide more experimental detail for Figure 4: how images are encoded and retrieved, how the number of preserved/emergent memories is counted, and what defines a preserved memory.
- Justify or ablate the VAE. Can LSR energy retrieve and generate meaningful memories directly in image space (e.g., for MNIST)? If not, why?
- Add a brief discussion in the introduction or conclusion about potential biological relevance—e.g., how different similarity kernels might exist across cortical circuits to balance memory and creativity.
- Evaluate other kernels (from Appendix B) in the DenseAM energy function to test if similar results arise with kernels like quartic, triweight, or cosine.

**Ethical Concerns:**

["NO or VERY MINOR ethics concerns only"]

**Final Justification:**

This original submission was already above the acceptance threshold. However, it lacked clarity for its experimental details and justifications. It also lacked experiments that can enhance the argument. During the rebuttal, the authors have convincingly resolved the problems I have raised in the initial comments (although due to the rebuttal format requirements, I can only imagine the visual results based on the authors' descriptions). I therefore raised the score. However, I still can't assess the mathematical derivations in the appendix due to personal expertise and bandwidth, so retained my confidence score.

**Limitations:**

Yes.

**Paper Formatting Concerns:**

No.

**Quality:**

3

**Strengths And Weaknesses:**

Strengths:
- The analogy between associative memory energies and KDEs is elegant and insightful. I enjoy reading how this paper is motivated by this analogy.
- The LSR energy allows for perfect memorization and emergence of novel, creative attractors. This provides a principled solution to the memorization–generalization trade-off in DenseAMs.
- Theorems and propositions throughout the paper clearly articulate properties of LSR energy, including capacity, convergence, and emergence. However, due to personal bandwidth and expertise I wasn't able to go through the whole appendix to examine the correctness of the proof. Therefore my recommendation on the theoretical part of this paper is conditional on confirmation by reviewers with appropriate background.

Weaknesses:
- The experiment details for figure 4 is insufficient. I'd love to see more information on the experimental pipeline, including how images are sampled, how the model is queried to generate (latent representations of) the images. It is also unclear to me why the VAE is needed here as denseAMs can directly perform memory retrieval in the image space. Maybe a sentence or so to justify the use of the VAE.
- Only one KDE kernel (Epanechnikov) is tested in the associative memory context. Appendix B lists other well-known KDE kernels—testing a few of these in the DenseAM setting would help clarify whether the observed behavior is unique to Epanechnikov or more general. (i understand this paper focuses on Epanechnikov kernel; however, testing other kernels, at least in the appendix, could strengthen the theoretical findingds of this paper)
- The biological/neuroscience implication of the findings is missing - the findings imply that the brain utilizes a unified computational principle (kernel/Hopfield networks) to perform both memorization and generalization. I feel this is an important implication biologically.

---

> ### Author Rebuttal · Authors · 2025-07-30
>
> We thank the reviewer for their time and effort in reviewing our paper. We are encouraged by the positive scores! Reviewers across the board have highlighted the following strengths of our work:
>
> 1. Regarding **novelty**, our work exploring the connections of KDE and Associative Memory is "novel" (Ua9r), "original" (TqXz), "elegant" (4hP4), and "insightful" (TqXz, 4hP4). It is a "conceptual breakthrough for the field of associative memories" (TqXz)
> 2. Regarding **rigor**, our work is "a principled solution to the memorization-generalization trade-off in DenseAMs" (4hP4) that "lays out a framework on studying emergent capabilities for future works" (Ua9r)
> 3. Regarding **quality**, the paper is "well-written and easy to follow" (TqXz) and "already above the acceptance threshold" (4hP4)
>
> Reviewers also noted that several discussions/experiments would further improve our work:
>
> 1. **Increase the scale of the current experiments** and confirm that the current conclusions hold (Ua9r)
> 2. **Clarify when emergent memory is meaningful** rather than harmful (TqXz)
> 3. **Discuss biological/neuroscience implications** (4hP4)
> 4. **Elaborate on the experimental details for Fig. 4**, the VAE real-data experiments (4hP4)
> 5. **Discuss emergence properties of other kernels** used for KDE (4hP4)
>
> We believe the feedback has increased the strength of our submission --- we summarize our changes in our responses to each reviewer. Unfortunately, this year's rebuttal format prevents us from sharing the revised submission and updated figures. We do our best to provide clarification and describe results in plain text.
>
> ---
>
> > I see the following potential opportunities that may promote this paper to a spotlight/oral one if the authors could include them
>
> Thank you for your vote of confidence! You have dangled a carrot for us; we hope to satisfy your reasonable requests with our answers below.
>
> > Provide more experimental detail for Figure 4.
>
> We flesh out the experimental methodology below for Fig. 4 below:
>
> **Designing the energy.** The DenseAM energies studied in this work are described by a matrix of stored patterns $\Xi$ and an inverse temperature hparam $\beta$.
>
> For *MNIST*, the experiment proceeds as follows: 24 random images from the MNIST training set are normalized to be $[0,1]$, rasterized into a vector of shape `(784,)`, and projected into a 10-dim latent space of a $\beta$-VAE trained according to the methods laid out in Appendix C.4 [L516 of our submission]. These latents become our stored patterns $\Xi \in \mathbb{R}^{24 \times 10}$ that we use to parameterize both the LSR energy and the LSE energy. We then use Alg. 2 to discover a $\beta$ value such that these stored patterns interact in the LSR energy to exhibit *emergence* ($\beta$ values that are too high will have no emergence, whereas $\beta$ values that are too small will have few emergent/no preserved memories) --- this $\beta$ value is also used to describe the LSE energy.
>
> For *TinyImagenet*, we randomly select 40 images from the dataset, each of shape `(C,H,W)=(3,64,64)`. These samples are passed these through a pretrained VAE called TAESD to produce latents that are of shape `(4,8,8)` which are then rasterized into vectors of shape `(256,)`. Thus, our stored pattern matrix $\Xi \in \mathbb{R}^{40 \times 256}$. Other details remain the same as MNIST.
>
> **Discovering local minima** Fig. 4 shows **all** memories for each system. Local minima are enumerated differently for the LSR and LSE energy. To clarify App. C.4 [L535 of our submission], LSR memories are discovered by filtering the set of "memory candidates" --- the set of centroids of stored patterns whose basins overlap --- to those whose energy gradient is zero. This method is explicitly described in Alg. 3. LSE memories are computed via gradient descent, initializing queries at the stored patterns and minimizing the energy until convergence. The set of unique fixed points from this inference is the complete set of LSE memories.
>
> We have clarified our submission with the details above.
>
> > Justify or ablate the VAE. Can LSR energy retrieve and generate meaningful memories directly in image space (e.g., for MNIST)? If not, why?
>
> This is an excellent question. We use the VAE for one primary reason that we discuss in the Limitations of our submission and which we clarify further in our response to Reviewer TqXz: *An emergent minima is exactly the centroid of some collection of stored patterns*.
>
> In MNIST pixel space, this results in memories that look like superpositions of digits: e.g., a "1" and a "7" displayed on top of each other, or a "2" that has been overlayed with a "5" and a "3" to create something like a blurry "8". In the structured latent spaces of deep models, we observe more interesting behavior. A "5" and a "9" can merge to become a "7", or two slanted "1"s will average out into a vertical "1", behaviors that do not occur when averaging in pixel-space. The "meaningfulness" of the emergent memory depends on the structure of the latent space.
>
> We wish we could communicate the idea of pixel-space superposition through a figure in this response. It is an elegant way to visualize how LSR energy works in high dimensions. We have updated our submission to include a discussion of emergence in pixel-space in the Appendix. We are grateful to the reviewer for raising this question.
>
> > Evaluate other kernels... to test if similar results arise
>
> Another excellent suggestion. Though our main focus was the Epanechnikov kernel, it turns out that the theoretical results in Prop. 2, Prop. 3, and Thm. 2 are *not* specific to this choice of kernel and indeed generalize to many kernels with *compact support*. This ensures the property that *local* averaging, rather than *global* averaging, can lead to emergence. The Epanechnikov kernel induces a localized interaction structure via the activation set $B(x) := \left[ \mu : |x - \xi_\mu|^2 < \frac{2}{\beta} \right]$, where emergent memories are centroids of subsets of stored patterns. Other kernels in App. B  will lead to different forms of the activation set $B(x)$ and thus different (and possibly more complex) averaging/emergence. For example, the Triangle kernel leads to a new set $B(x) := \left[ \mu : |x - \xi_\mu| < \frac{2}{\beta} \right]$. Results in Prop. 2, Prop. 3, and Thm. 2 are expected to hold with corresponding changes. Interestingly, all these kernels exhibit exponential memory capacity when viewed from the perspective of Associative Memory.
>
> To demonstrate this behavior, we recreate Fig. 1 (left) to show 1D merging of two energy basins for every kernel in App. B (i.e., Triangle, Uniform, Cosine, Quartic, Triweight, and Tricube) at different $\beta$ (bandwidth). This new figure shows what "emergence" looks like for each of them. We describe their behavior below.
>
> 1. **Triangle**. The Triangle kernel exhibits very interesting emergence behavior. A perfectly flat energy manifold is formed where two or more basins overlap. If the energy of this manifold is lower than the energy of the stored patterns, the stored patterns "merge" and are no longer retrievable. If the energy of the manifold is higher, both original patterns are preserved and we observe a locally emergent *manifold* of memory.
> 2. **Uniform**. A Uniform kernel produces an energy landscape that is not continuous and arguably impractical for associative memory. Emergent minima have exclusively lower energy than the stored patterns. Except for discontinuities, the entire energy landscape is flat.
> 3. **Cosine**. The Cosine kernel looks remarkably like the Epanechnikov kernel, and its emergence properties are almost identical to that of LSR. It appears that Cosine-emergent memories have slightly higher energy than their Epanechnikov counterpart.
> 4. **Quartic**. The Quartic kernel produces smoother energy landscapes than the Epanechnikov does. Overlapping basins does not guarantee emergent minima, but carefully tuning $\beta$ discovers emergence. This behavior is an interesting departure from the previous kernels that will need to be studied more carefully.
> 5. **Triweight**. The Triweight kernel behaves like the quartic kernel in that emergence is difficult to find. The closest we were able to discover is a nearly flat manifold connecting the two stored patterns where everything is at the same, flat energy.
> 6. **Tricube**. Behaves like the Cosine kernel, but the energy is noticeably smoother. Tricube's emergent minima have higher energy than cosine's at the same bandwidth.
>
> We include this discussion alongside a figure describing their behavior in our updated submission.
>
> > Add a brief discussion in the introduction or conclusion about potential biological relevance
>
> Dense Associative Memories permit a standard mapping onto "biological" networks by introducing auxiliary hidden neurons following the method of (Krotov & Hopfield, ICLR 2021 - Ref [6]). Thus, the $d$ feature neurons in both LSE and LSR models can be augmented with auxiliary $M$ hidden neurons, resulting in a model with only pair-wise neuron interactions and bipartite connectivity between feature and hidden layers. This augmented model is more biological compared to the model considered in the submission. The proposed model defined by the energy Eq. (3) and the corresponding update rule can be obtained by integrating out these auxiliary hidden neurons, which means LSR energy will still have the same emergent behavior. The augmented model will still have some un-biological aspects because of the weight symmetry between forward and backward projections, which is necessary to ensure that the network has a global energy function. This discussion has been added to the revised version of the paper.
>
> ---
>
> We thank you again for your interest in this work and your helpful feedback! If our response above satisfies your original concerns, we would greatly appreciate an increase of score to reflect your increased confidence in our work.

---

> ### Comment · Reviewer_4hP4 · 2025-08-03
>
> I would like to thank the authors for their detailed reply to my comments and the added experiments. It's a pity that in no way can authors demonstrate any visual results this year during rebuttal. For certain fields in neurips this is prohibiting effective communications between reviewers and authors.
>
> I appreciate the authors' justification for the usage of VAE. I totally agree that the superposition is a problem for AM in pixel space, but should be a desirable property in latent space. For the added experiments with other kernels, I can only choose to trust the authors for their descriptions of the visual results. However, I hope this caveat of rebuttal format can be addressed in future NeurIPS. I will thus raise my score.

---

### Official Review · Reviewer_TqXz · 2025-07-03

**Clarity:** 4
**Significance:** 4
**Originality:** 4
**Rating:** 5
**Confidence:** 4

**Summary:**

This paper introduces a novel energy function for DenseAMs, termed LSR, which is derived from the principles of optimal Kernel Density Estimation (KDE). By establishing a connection between AM energy functions and KDE, the authors propose using a ReLU-based separation function, analogous to the statistically optimal Epanechnikov kernel, as an alternative to the commonly used LSE formulation (which corresponds to a Gaussian kernel). The primary contribution of this work is to demonstrate that the LSR energy function resolves the fundamental trade-off between perfect memorization and generalization. The authors provide theoretical proofs and empirical evidence showing that LSR can achieve exponential storage capacity with *exact* retrieval of all stored patterns, while simultaneously creating a vast number of novel "emergent" local minima. These emergent memories, which arise as centroids of overlapping subsets of stored patterns, are a unique feature not present in LSE-based models under the condition of perfect recall. Experiments on synthetic data and latent spaces of image datasets validate that LSR produces a significantly richer and more diverse set of high-quality memories than LSE, showcasing its potential for both high-capacity storage and generative tasks.

**Questions:**

Typo: “textvol” in Proposition 3

**Ethical Concerns:**

["NO or VERY MINOR ethics concerns only"]

**Final Justification:**

I have carefully read the author's responses to my comments and the feedback from the other reviewers. The author has addressed all of my questions and concerns effectively. I am now confident in supporting the publication of this manuscript and will maintain my initial high score.

**Limitations:**

yes

**Quality:**

4

**Strengths And Weaknesses:**

**Strengths:**

This is a strong paper with great contributions. The central idea of leveraging insights from optimal KDE to design a new energy function for associative memories is original and insightful. The paper's main finding is a conceptual breakthrough for the field of associative memories. The phenomenon of "global emergence", where a model can simultaneously achieve exact memorization and generate a vast number of novel patterns, is attractive. It provides a more profound understanding of the memorization-generalization trade-off AMs. The paper is well-written and easy to follow. The motivation is clearly articulated, logically flowing from the limitations of existing AMs to the KDE-inspired solution.

**Weaknesses:**

The emergent memories are mechanistically simple, being the centroids of subsets of stored patterns. The generative capability is therefore predictable and less expressive. The authors do not justify why the emergent memory is meaningful rather than harmful for real tasks. For example, while the paper regards emergent memories as a desirable and controlled form of generalization, these novel patterns are not explicitly present in the stored data. It would be beneficial to discuss the relationship between the emergent memory and model hallucination.

---

> ### Author Rebuttal · Authors · 2025-07-28
>
> We thank the reviewer for their time and effort in reviewing our paper. We are encouraged by the positive scores! Reviewers across the board have highlighted the following strengths of our work:
>
> 1. Regarding **novelty**, our work exploring the connections of KDE and Associative Memory is "novel" (Ua9r), "original" (TqXz), "elegant" (4hP4), and "insightful" (TqXz, 4hP4). It is a "conceptual breakthrough for the field of associative memories" (TqXz)
> 2. Regarding **rigor**, our work is "a principled solution to the memorization-generalization trade-off in DenseAMs" (4hP4) that "lays out a framework on studying emergent capabilities for future works" (Ua9r)
> 3. Regarding **quality**, the paper is "well-written and easy to follow" (TqXz) and "already above the acceptance threshold" (4hP4)
>
> Reviewers also noted that several discussions/experiments would further improve our work:
>
> 1. **Increase the scale of the current experiments** and confirm that the current conclusions hold (Ua9r)
> 2. **Clarify when emergent memory is meaningful** rather than harmful (TqXz)
> 3. **Discuss biological/neuroscience implications** (4hP4)
> 4. **Elaborate on the experimental details for Fig. 4**, the VAE real-data experiments (4hP4)
> 5. **Discuss emergence properties of other kernels** used for KDE (4hP4)
>
> We believe the reviewer feedback has increased the strength of our submission, whose changes we summarize in our responses to each reviewer. Unfortunately, this year's NeurIPS rebuttal format prevents us from sharing the revised submission and updated figures. We do our best to provide clarification and describe results in plain text.
>
> ---
>
> ---
>
> > ...justify why the emergent memory is meaningful rather than harmful for real tasks.
>
> This is a really good question, and one that is hard to answer without nuance. Indeed, it is difficult to know if a novel generation is "good" or "bad" --- it is more helpful to say if it is *useful* or not for a particular task.
>
> For example, in Fig. 3 (right) we show that the LSR energy approximates an unknown PDF better than LSE's energy while simultaneously generating diverse samples. This is a *desirable* behavior of emergence, since high log-likelihood samples from the LSE energy are quite homogeneous, where 500 initial queries converge to the same ~10 memories. On the other hand, consider the novel memories from Tiny Imagenet in Fig. 4. Though plausible, many of the novel generations are blurry and would be considered *undesirable* from the perspective of an image generation model.
>
> We include a more fleshed out version of this response regarding meaningful vs. harmful emergent memories in our camera-ready submission.
>
> > ...discuss the relationship between the emergent memory and model hallucination.
>
> The reviewer raises an intriguing point about the connection between emergent memories and hallucination. In everyday use of LLMs, we call it a *hallucination* when the model confidently produces an incorrect fact, especially when we know the model is capable of producing the correct fact in other settings. But to reason about hallucination in the context of emergent memories, we must imagine a setting where the model has an explicit energy function and a known set of stored “facts.”
>
> This is the setting of Associative Memory, where the outputs of our model are always energy minima (memories). A "hallucination" can occur when two or more stored facts interfere, leading the system to settle into an emergent minimum that looks meaningful (i.e., it lies near the "factual manifold" that generated our stored patterns) but does not correspond to any true, stored fact. Emergent memories are precisely these "interference minima" when they coexist with the original stored patterns. By definition, they are not stored facts, yet they can resemble meaningful combinations of them --- making them the *memory‑system analog of hallucinations*.
>
> This said, we can imagine a highly structured latent space where interpolations between facts are facts themselves. For example, if stored patterns are facts representing logical statements, and the local average of two or more facts in a feature space corresponds to an `OR` operation, then that emergent memory would also be factual. But in general, emergent memories illustrate exactly how a system can produce plausible‑looking outputs that are not true to the stored data.
>
> We believe this is an interesting discussion and provide a brief note of it in the appendix of our submission. Full discussion of "hallucination from the perspective of emergence" would require more rigorous definitions and exploration that is outside the scope of this work, but we share your interest in this relationship!
>
> > The emergent memories are mechanistically simple, being the centroids of subsets of stored patterns
>
> Indeed, the emergent memories are mechanistically simple. But the simplicity of emergent minima in the LSR energy is also a benefit. We can take advantage of this simplicity to develop efficient inference that allows us to converge to the *exact* fixed point in as little as a single step (Alg. 1). Meanwhile, one-step retrievals on the LSE energy converge to approximate fixed points [3]. We can also take advantage of the simplicity to efficiently predict candidate emergent memories without needing to explore the entire search space.
>
> We also point out that, with enough stored patterns at the right $\beta$, local linear interpolations can approximate globally curved manifolds. That is, by placing memories on the centroids of linear interpolations between any 2 (or 3, 4, etc.) data points, the LSR energy can be used to approximate arbitrary manifolds. Even deep ReLU networks approximate complex functions by learning a many linear approximations to curves. We elaborate on this in our camera-ready submission.
>
> > Typo: “textvol” in Proposition 3
>
> Thank you for the catch! We have fixed this in our submission
>
> [3] Ramsauer et al., "Hopfield Networks is All You Need". ICLR 2021
>
> ---
>
> We thank you again for your insightful feedback and questions. If our response above satisfies the reviewer's original concerns, we would greatly appreciate an increase of score to reflect your increased confidence in our work.

---

> > ### Comment · Reviewer_TqXz · 2025-08-03
> >
> > I would like to thank the authors for their detailed reply to my comments! The meaning of emergent memory is worth discussing in the camera-ready version. Since I have given a high score in the initial states, I will keep the score.

---

### Official Review · Reviewer_Ua9r · 2025-07-17

**Clarity:** 4
**Significance:** 3
**Originality:** 3
**Rating:** 4
**Confidence:** 3

**Summary:**

The paper explores energy-based associative memory networks. A notable prior work is the LSE energy function, which yields exponential memory capacity. However, it typically faces a trade-off between perfect memorization of stored patterns and the generalization to novel patterns. This paper proposes a new energy function log-sum-relu (LSR) inspired by the connection between energy functions and probability density functions. Through some exploratory experiments, the paper shows promising demonstrations for exact memorization of all original patterns, and simultaneously generates new, meaningful emergent memories.

**Questions:**

* On the use of ReLU and its relevance to the modern deep networks. ReLU was proposed in part to facilitate gradient flows and avoid the gradient vanishing problem. Does ReLU play a similar role in LSR? How about studying other more advanced activation functions such as GeLU and GeGLU?

* It may be non-trivial to stack multiple LSR to form deep networks of memories. I am wondering the potential of formulating and implementing this.

**Ethical Concerns:**

["NO or VERY MINOR ethics concerns only"]

**Final Justification:**

My concerns have been well addressed. I decide to keep my positive rating.

**Limitations:**

Yes

**Quality:**

3

**Strengths And Weaknesses:**

Strengths:

* The paper provides a novel view regarding the connection between energy functions and KDE.

* Theoretical contributions are made especially for generating novel emergent memories. This also lays out a framework on studying emergent capabilities for future works.

Weakness:

* The scale of the current experiment results is too small. For results shown in section 4.3, only a handful of less than 100 images are used for memorization. It remains unclear if the training images scale to thousands or millions, the current conclusion might not be hold.

* The paper now lacks quantitative experimental evaluations. While theoretical contributions and empirical demonstrations are nice, it is beneficial to follow LSE paper for realisitic quantitive evaluations, such as on classification problems.

---

> ### Author Rebuttal · Authors · 2025-07-28
>
> We thank the reviewer for their time and effort in reviewing our paper. We are encouraged by the positive scores! Reviewers across the board have highlighted the following strengths of our work:
>
> 1. Regarding **novelty**, our work exploring the connections of KDE and Associative Memory is "novel" (Ua9r), "original" (TqXz), "elegant" (4hP4), and "insightful" (TqXz, 4hP4). It is a "conceptual breakthrough for the field of associative memories" (TqXz)
> 2. Regarding **rigor**, our work is "a principled solution to the memorization-generalization trade-off in DenseAMs" (4hP4) that "lays out a framework on studying emergent capabilities for future works" (Ua9r)
> 3. Regarding **quality**, the paper is "well-written and easy to follow" (TqXz) and "already above the acceptance threshold" (4hP4)
>
> Reviewers also noted that several discussions/experiments would further improve our work:
>
> 1. **Increase the scale of the current experiments** and confirm that the current conclusions hold (Ua9r)
> 2. **Clarify when emergent memory is meaningful** rather than harmful (TqXz)
> 3. **Discuss biological/neuroscience implications** (4hP4)
> 4. **Elaborate on the experimental details for Fig. 4**, the VAE real-data experiments (4hP4)
> 5. **Discuss emergence properties of other kernels** used for KDE (4hP4)
>
> We believe the reviewer feedback has increased the strength of our submission, whose changes we summarize in our responses to each reviewer. Unfortunately, this year's NeurIPS rebuttal format prevents us from sharing the revised submission and updated figures. We do our best to provide clarification and describe results in plain text.
>
> ---
>
> >  It remains unclear if the training images scale to thousands or millions, the current conclusion might not be hold.
>
> It seems like the reviewer is requesting empirical confirmation of Thm. 2 in the regime of large data (please correct us if our interpretation of "current conclusion" is amiss). To satisfy this, we design an experiment to store all 60k rasterized MNIST training images into the LSR DenseAM. In this large data regime, we show multiple examples of novel images forming between two or more stored patterns that are still retrievable. This confirms the primary claim of the paper: that LSR energy can create *emergent memories* --- novel energy minima co-existing with energy minima of the original stored patterns --- at any scale of data.
>
> Note that our current submission focuses on quantifying and studying the complete set of *emergent memories* in regimes that we could feasibly compute. It becomes computationally infeasible to enumerate all the emergent memories in the regime of high-dimensional spaces $d$ and a large number of stored patterns $M$. For instance, consider the largest experimental setup of Fig. 3 (Left), where $d=32$ and $M=25$. To enumerate all possible emergent minima, we need to check the energy gradient of the centroids of all possible subsets of the 25 stored patterns --- a combinatorial search problem over $2^M$ possible centroids per each tested $\beta$. The real-data experiments in Fig. 4 are even larger, where we store $M=40$ patterns of dimension $d=256$, requiring a search over $2^{40}$ possible centroids. We can optimize this search using e.g., Alg. 3 in Appendix C, but for large $M$ (like the complete MNIST training set), even this algorithm would computationally fail at enumerating all minima for interesting $\beta$.
>
> > ReLU was proposed in part to facilitate gradient flows and avoid the gradient vanishing problem. Does ReLU play a similar role in LSR? How about studying other more advanced activation functions such as GeLU and GeGLU?
>
> ReLU is a particularly interesting function since it is prevalent in many modern architectures, though our inspiration for ReLU in the LSR energy comes primarily from KDE literature. ReLU does indeed help prevent vanishing gradients during memory recall in Associative Memories compared to e.g., the sigmoid function, where gradients are tiny for large positive or negative inputs to the function. Just note that the "vanishing gradient" problem in our work refers to gradients vanishing during *inference* (energy minimization w.r.t. the inputs), which is different from the vanishing gradient observed during traditional AI *training* (loss minimization w.r.t. the parameters).
>
> Despite their similarity to the ReLU, neither the GeLU or GeGLU are valid activations for our architecture for the following reasons:
>
> 1. **Ge[G]LU violates a core tenet of Associative Memory** because it is not a monotonic function (i.e., its Jacobian is not positive definite), which means the energy is *not* guaranteed to monotonically decrease with gradient descent. We are happy to elaborate on this more if you have additional questions.
> 2. **Ge[G]LU is not a proper kernel** since it has regions where it dips negative. This behavior of "negative probability density" usually has no clear meaning and application in KDE.
>
> > It is beneficial to follow LSE paper for realisitic quantitive evaluations, such as on classification problems
>
> We appreciate the reviewer’s suggestion and agree that classification experiments, as in [2], can provide a valuable benchmark. In this work, however, we deliberately focus on a canonical evaluation for the "memory" aspect of DenseAMs --- the storage capacity and retrieval quality of stored patterns. Thus, our experiments validate LSR energy's ability to store patterns (Fig. 3 left) and the quality of those stored patterns (quantitatively in Fig. 3, right, and qualitatively in Fig. 4) relative to the LSE baseline. These experiments do not require training the stored patterns themselves.
>
> It is an exciting research direction to incorporate training procedures such as backpropagation or contrastive learning into the LSR energy, and we expect both techniques to produce interesting research challenges of their own. We believe that training these architectures is an orthogonal research dimension to the contributions of this work, one that would be best explored in a dedicated follow‑up paper.
>
> > It may be non-trivial to stack multiple LSR to form deep networks of memories. I am wondering the potential of formulating and implementing this.
>
> This is a fantastic observation. Indeed it is possible, though as you pointed out the mathematics are non-trivial. Following the methods of "Hierarchical Associative Memory" [1], we can make our energy function deeper by introducing both latent variables and new parameter matrices. Each parameter matrix is coupled to a latent variable and will store patterns observed in those latent variables. To extract information from these matrices, we can use an Epanechnikov kernel (or any other kernel discussed in Appendix B) to compute similarities between a "query" latent and its parameter matrix. This results in a new energy that represents a deep, intricately connected network of memories.
>
> However, this architecture is a large departure from the fundamental discussion in our submission and we will leave complete evaluation of this "Hierarchical Epanechnikov AM" to future work.
>
> [1] Krotov, "Hierarchical Associative Memory". ArXiv preprint, 2021
>
> [2] Krotov \& Hopfield, "Dense Associative Memory". NeurIPS, 2016
>
>
> ---
>
> We thank you again for your insightful feedback and questions! If our response above satisfies the reviewer's original concerns, we would greatly appreciate an increase of score to reflect your increased confidence in our work.

---

### Decision · Program_Chairs · 2025-09-17

**Decision:**

Accept (spotlight)

**Comment:**

The paper proposes a form of associative memory networks represented by a new energy function, log-sum-ReLU (LSR), inspired by the connection between energy functions and probability density functions. All reviewers are positive about the submission. Leveraging insights from optimal KDE to design a new energy function for associative memories is considered both original and insightful, and the ability to achieve exact memorization while generating a large number of novel patterns is particularly attractive. Although some concerns were raised regarding the experimental results, these were satisfactorily addressed during the rebuttal.

Overall, the paper makes a strong contribution by advancing our understanding of the fundamentals of generalization. It may be able to attract broad interest within the machine learning community. The AC therefore recommends clear acceptance with a spotlight presentation.